



# Effects on the Czech Lands of the 1815 eruption of Mount Tambora: responses, impacts and comparison with the Lakagígar eruption of 1783

Rudolf Brázdil[1,2], Ladislava Řezníčková[1,2], Hubert Valášek[1,3], Lukáš Dolák[1,2], Oldřich Kotyza[4]

[1]Institute of Geography, Masaryk University, Brno, Czech Republic
[2]Global Change Research Institute, Czech Academy of Sciences, Brno, Czech Republic
[3]Moravian Land Archives, Brno, Czech Republic
[4]Regional Museum, Litoměřice, Czech Republic

*Correspondence to*: R. Brázdil (brazdil@sci.muni.cz)

**Abstract.** The eruption of Mount Tambora in Indonesia in 1815 was one of the most powerful of its kind in recorded
history. This contribution addresses climatic responses to it, the post-eruption weather, and its impacts on human life in
the Czech Lands. The climatic effects are evaluated in terms of air temperature and precipitation on the basis of long-
term homogenised series from the Prague-Klementinum and Brno meteorological stations, and mean Czech series in the
short term (1810–1820) and long-term (1800–2010). This analysis is complemented by other climatic and
environmental data derived from rich documentary evidence. Czech documentary sources make no direct mention of
the Tambora eruption, neither do they relate any particular weather phenomena to it, but they record extremely cold and
wet summers for 1815 and 1816 (the "Year Without a Summer") that contributed to bad grain harvests and widespread
grain price increases in 1817. Possible reasons for the cold summers in the first decade of the 19th century cited in the
contemporary press included comets, sunspot activity, long-term cooling and finally – as late as 1817 – earthquakes
with volcanic eruptions. Here, the Tambora event is compared with the 1783 eruption of Lakagígar in Iceland, with its
clearly-pronounced post-volcanic effects on the weather in central Europe (dry fog, heavy thunderstorms, optical
phenomena) and the occurrence of significant cold temperature anomalies in winter 1783/84, spring 1784 and the
summer and autumn of 1785. These appeared clearly in central European series, Prague-Klementinum included.
Comparison of the two eruptions shows that the effects of the Lakagígar eruption in the Czech Lands were
climatologically stronger those of the Tambora eruption, while the opposite held for societal responses.

Key words: documentary data – climate – Tambora eruption – Lakagígar eruption – human impacts – Czech Lands

## 1 Introduction

Violent tropical volcanic eruptions, transporting large quantities of particles into the lower stratosphere, give rise to
decreases in temperatures in the troposphere, which cools for two or three subsequent years in response to strongly
enhanced back-scattering of incoming solar radiation (Robock and Mao, 1995; Briffa et al., 1998; Robock, 2000; Jones
et al., 2004; Písek and Brázdil, 2006; Timmreck, 2012; Lacis, 2015; LeGrande and Anchukaitis, 2015). Camuffo and
Enzi (1995) studied the occurrence of clouds of volcanic aerosols in Italy over the past seven centuries with particular
attention to the accompanying effect of "dry fog". Volcanic cooling effects are best expressed in temperature series
averaged for a large area after significant tropical volcanic eruptions (Sear et al., 1987; Bradley, 1988; Briffa et al.,
1998; Sigl et al., 2015). For example, Fischer et al. (2007) analysed winter and summer temperature signals in Europe
following 15 major tropical volcanic eruptions and found significant summer cooling on a continental scale and
somewhat drier conditions over central Europe. Literature addressing volcanic effects on precipitation is more sparse
(Gillett et al., 2004). Wegmann et al. (2014) analysed 14 tropical eruptions and found an increase of summer





precipitation in south-central Europe and a reduction of the Asian and African summer monsoons in first post-eruption years. Weaker monsoon circulations attenuate the northern element of the Hadley Cell and influence atmospheric circulation over the Atlantic-European sector, contributing to higher precipitation totals.

A great deal of literature has been devoted to analysis of the climatological and environmental effects of the Tambora eruption. The subsequent year of 1816 has been termed the "Year Without a Summer" (see e.g. Stommel and Stommel, 1983; Harington, 1992; Vupputuri, 1992; Habegger, 1997; Oppenheimer, 2003; Bodenmann et al., 2011; Klingaman and Klingaman, 2013; Brugnara et al., 2015; Luterbacher and Pfister, 2015). Kužić (2007) investigated the effects in Croatia of an unidentified eruption in 1809 and the 1815 Tambora event. Trigo et al. (2009) studied Tambora impacts in Iberia using both documentary and instrumental data. Lee and MacKenzie (2010), referring to a farming

diary from north-west England that held weather entries for 1815–1829, found significant climate anomalies for the two years following the Tambora eruption. Auchmann et al. (2012) paid particular attention to the weather and climate of the 1816 summer for Geneva (Switzerland). Cole-Dai et al. (2009) held Tambora, with a further unidentified tropical eruption in 1809, responsible for the bitter 1810–1819 period, probably the coldest decade of the last 500 years or longer. However, Guevara-Murua et al. (2014) attributed the unidentified 1809 eruption to late November/early

December 1808, as the second most explosive sulphur-dioxide-rich volcanic eruption for the last two centuries. Büntgen et al. (2015) identified the 1810s as coolest summer decade for the last three centuries in central Europe, basing this conclusion on tree-rings from 565 samples of Swiss stone pine (*Pinus cembra*) from high-elevation sites in the Slovak Tatra Mountains and the Austrian Alps. Briffa and Jones (1992) classified just the summer of 1816 as extreme in that particular decade in Europe.

Considerable attention has also been dedicated to the climatic and environmental responses to other specific volcanic events, in particular the Lakagígar (Laki) eruption of 1783 in Iceland (Thordarson and Self, 1993, 2003), which was associated with high mortality in western Europe (Grattan et al., 2003, 2005; Witham and Oppenheimer, 2005). Vasold (2004) reported on summer 1783, followed by a severe winter and disastrous flood in February 1784, in Germany. Brázdil et al. (2010a) described extreme floods in Europe following the first, very severe, winter after the

eruption (1783/1784).

This contribution aims to provide a comprehensive description of climatic and environmental responses to the Tambora 1815 eruption for the Czech Lands (recently the Czech Republic), contrasted with the known effects of the Lakagígar 1783 eruption. Section 2 addresses temperature and precipitation instrumental series, weather-related documentary data and the socio-economic data used in this study. Section 3 presents methods used for the study of

short-term and long-term responses. Section 4 gives a full description of the climatic and environmental consequences of the two eruptions in the Czech Lands. A comparison is drawn and the broader context of their impacts is discussed in Section 5. The final section summarises the most important findings.

## 2 Data

### 2.1 Instrumental data

The climatological analysis herein is based on monthly, seasonal and annual temperature and precipitation series for the Czech Lands (Fig. 1). Of ten homogenised series of monthly temperatures and 14 series of precipitation totals, only two cover the time of Tambora eruption: Prague-Klementinum (temperatures 1775–2015, precipitation 1804–2015) and Brno (1800–2015 and 1803–2015, respectively) (Brázdil et al., 2012a, 2012b; extended to 2011–2015). They are

complemented by mean Czech areal temperature (1800–2015) and precipitation (1804–2015) series calculated from these stations (Brázdil et al., 2012a, 2012b; extended to 2011–2015). Two additional temperature series are employed: monthly temperatures from the meteorological measurements of František Jindřich Jakub Kreybich of Žitenice (1787–





1829) (Brázdil et al., 2007) and a temperature reconstruction for central Europe derived from documentary-based temperature indices from Germany, Switzerland and the Czech Lands and 11 homogenised long-term stations in these countries and Austria for AD 1500–2007 (Dobrovolný et al., 2010).

**2.2 Documentary data**

The pre-instrumental and early-instrumental period of meteorological observations in the Czech Lands is well covered by documentary evidence that contains information about weather and related phenomena. It occurs in a number of data sources, including annals, chronicles, memoirs, diaries, newspapers, financial records, songs, letters, epigraphic records, and others (Brázdil et al., 2005b, 2010b). The following daily visual weather records are of particular importance within

this broad and plentiful body of documentary evidence:

1) Reverend Karel Bernard Hein of Hodonice (south-western Moravia), observations 1780–1789 (Brázdil et al., 2003);

2) Reverend Šimon Hausner of Buchlovice (south-eastern Moravia), observations 1803–1831 (S4).

The notes that accompany instrumental observations by Kreybich in Žitenice are also very valuable (Brázdil et al., 2007; S1–S3), together with detailed weather records kept by Anton Lehmann, a teacher in Noviny pod Ralskem

(S6) and observations by Antonín Strnad and Alois David, the third and fourth directors of the Prague-Klementinum observatory (Poznámky, 1977).

The editions of newspapers published in Prague (*Prager Zeitung*), Brno (*Brünner Zeitung*) and Vienna (*Wiener Zeitung*) covering the post-Tambora years were also systematically scrutinised, although weather information appears relatively rarely in their pages with respect to descriptions of events in the Czech Lands or Austria, while such stories

from other parts of Europe or North America clearly prevail.

**3 Methods**

The climatic effects of the volcanic eruption are expressed in the short-term and long-term contexts. In the short-term, climatic patterns related to the eruption are described over a ten-year period to avoid the possible influence of a strong

trend. The month of the eruption is taken as month zero. The mean temperature for each month was calculated using temperature data from five years prior to the eruption. Each monthly average temperature for five years before and after the eruption was then expressed as a departure from the calculated mean value. The same approach was applied to series of precipitation totals. For the long-term context, the eruption year and two subsequent years were characterised by their order and magnitude in the whole series shown in increasing (temperatures) or decreasing (precipitation) order.


**4 Results**

**4.1 Climatic, weather and social responses to the Tambora eruption**

The volcanic eruption of Tambora (Lesser Sunda Islands, Indonesia) in April 1815, classified at VEI-7, is among the most powerful of its kind recorded. The following year of 1816 has been characterized as the "Year Without a Summer"

(see e.g. Stommel and Stommel, 1983; Stothers, 1984; Harington, 1992; Vupputuri, 1992; Oppenheimer, 2003). During the Tambora eruption, around 60 Tg of $SO_2$ were thrown into the stratosphere, where the $SO_2$ oxidized to sulphate aerosols (Self et al., 2004; Kandlbauer and Sparks, 2014).

**4.1.1 Climatic responses**

Fig. 2 shows seasonal temperature anomalies for the Prague-Klementinum, Žitenice and Brno stations and for mean series for the Czech Lands and central Europe. These are expressed with respect to the 5-year period pre-eruption. Cooling, as indicated by negative anomalies, is already evident in the summer and autumn of 1815 and, after the





slightly positive winter of 1815/1816 temperature anomaly it continued for the rest of 1816. After a very mild winter of 1816/1817 (with the exception of Brno, the mildest in the 1811–1820 period), negative anomalies occurred, especially in spring with the strongest negative anomaly (stronger than in summer 1816). Autumn 1817 also exhibited a negative anomaly. However, it also follows from Fig. 2 that a cooler period was already in process from spring 1812 to autumn

1814, interrupted by slightly positive anomalies in spring 1813 at two Bohemian stations (Prague-Klementinum, Žitenice), while warm patterns prevailed in 1811.

Among monthly temperature anomalies, April 1817 is worthy of mention, fluctuating between –4.2°C and –4.8°C for the five series studied. Other very cold months included October 1817, December and July 1815 (*cf.* Fig. 8). A considerable drop in differences between mean winter and summer temperatures in 1815 demonstrates a clear

reduction of seasonality after the Tambora eruption for all five series (Fig. 3). This is related to the fact that tropical eruptions induce a positive phase in NAO circulation over Europe in the first and second years post-eruption, leading to winter warming on the one hand and summer radiative cooling due to volcanic aerosols on the other (Fischer et al., 2007).

Seasonal precipitation anomalies in the Prague-Klementinum, Brno and Czech Lands series (Fig. 4) exhibited

positive anomalies in both summer 1815 (particularly June and partly August) and 1816 (mainly June), with the first-mentioned particularly rainy. Another clear but negative anomaly occurred in autumn 1817, while the remainder of the 1815–1817 seasons showed somewhat smaller, or even opposite, anomalies. These may be attributed either to natural spatial differences in precipitation totals between two stations distant from one another, or to weaknesses in the homogenisation of precipitation series (lower spatial correlations and lack of stations for calculation of reference series

for the past) (Brázdil et al., 2012a).

### 4.1.2 Post-volcanic weather and impacts on society

**The year 1815**

Šimon Hausner, a reverend, kept daily weather records for Buchlovice. He mentions a rather cold May 1815 with more frequent rain and frosts on 29–30 May. Further, he characterises June, after some early fine days, as a windy and rainy month. July weather was variable, with frequent rain, strong winds, and cold mornings and evenings; the whole month was somewhat cooler than usual. August was rainy, with the exception of a few days, often with "torrents of water" [*Wassergüsse*]. Haymaking and the grain harvest (particularly wheat) took place in rainy weather. If two days were fine,

it then rained again for two days. The wine vintage was bad for the third year (S4). František Jindřich Jakub Kreybich, a cleric in Žitenice, speaks of the leaves on fruit trees entirely eaten away by caterpillars in May. Moreover, at the beginning of the following month, the wheat and some of the rye were infested with rust. Periods of rain in July–August complicated the harvest at higher altitudes in particular, where all the hay rotted (S1). A message from Litoměřice dated 9 August reports a flood lasting eight days on the River Elbe after five weeks of rainy periods. The water rose to a level

of two feet [*c.* 65 cm] under the bridge, so the structure survived, but grain, vegetable and other field crops were damaged (Katzerowsky, 1895). In a similar vein, Kreybich reports a flood on the Elbe for 10–14 August with extensive damage to agricultural crops (S1). A flood on the River Vltava, reported for 9–10 August for Prague, inundated fields and damaged crops (Brázdil et al., 2005a). Flood damage to fields tied to the aristocracy was reported around the Bečva River at Troubky (Brázdil and Kirchner, 2007).

The wet, cold summer gave way at the end of August to a very dry, cold autumn in 1815, confirmed by sources from Bohemia (S1) and Moravia (S4), and clearly documented by lower monthly numbers for precipitation days (Fig. 5). The grape harvest was below average in terms of both quality and quantity (Katzerowsky, 1895), there was no fruit



and the potato yield was bad (Bachmann, 1911). Frosts set in from 7 December at Buchlovice (S4), but on 1 January 1816 the ice-floes had dispersed from the River Elbe at Roudnice nad Labem and Litoměřice (S1).

**The year 1816**

Hausner describes the two winter months of 1816 in Buchlovice as: January – relatively cold weather to mid-January, mild with rain afterwards; February – variable with deep frosts on the one hand and periods of thaw on the other (S4). Kreybich records for Žitenice describe January as mild and February as much colder with the Elbe and Ohře rivers frozen from 8 to 20 February. The ice was definitely gone by 8–9 March (S2). Lehmann reports a 3/4-ell [*c*. 58-cm]-thick crust of ice on some fields in Noviny pod Ralskem (S6). Frosty weather prevailed in March with blizzards from 26 to 31 March. April was cold and dry, with no heavy rain (S4). Other Czech documentary sources report 1816 as

particularly cold and wet, with bad harvests and rising prices of all products. Around Nové Město na Moravě in the Bohemian-Moravian Highlands, lingering snow cover hampered the spring sowing, which started as late as 15 May (Trnka, 1912). Václav Jan Mašek of Řenče, who kept records, writes: "[...] *started to rain on St. Medard's day* [8 June] *and* [continued to do so] *for eight weeks, such that for this entire time one day in the week without rain was rare;*

*around St. John's* [24 June], *when the hay was harvested, God granted a few fine days* [...] *All the grain was saturated, it was too wet to dig the potatoes and from this* [situation] *it followed that the yield was bad, prices rose terribly high and hunger* [appeared]" (Urban, 1999). Šimon Hausner's monthly summary for Buchlovice describes the summer as: June – rainy, very windy, cold, little warmth; July – little warmth, mostly rain and strong winds, people driven by poverty to start harvesting early; August – except for a few days, cold and wet weather, harvest continued long time

(S4). Kreybich reports cold and rainy weather from May onwards, for the whole summer up to September. For the summer months, he makes particular mention of a number of unusually dense fogs and damaging thunderstorms. It rained for 191 days of the year at Žitenice (S2). Records kept by Martin and František Novák in Dřínov report a bad grain harvest (frost damage in May, especially to the rye), almost no fruit, wetness and rainy periods. The wheat was harvested very late, around 21 September. Barley was added to bread mixes, but it was not long before nearly every

possible substitute came into use – oats, vetch, peas, potato, and acorns are mentioned. Many farmers fell into debt (Robek, 1974). The "Book of Memory" for the school in Chrást even mentions that "*in many small villages, people prepare grass scalded in hot milk for food, and* [also] *eat bran*" (Anonymous, 1919).

The subsequent wet autumn of 1816 saw delays to the bean harvest and autumn sowing; winter wheat was sown late, even delayed to 5 November around Boskovice (S5). Reports from Olešnice indicate that low rye yields

meant that some farmers had to use anything available for new sowing (Paměti starých písmáků moravských, 1916). No wine was available in Litoměřice (Katzerowsky, 1895). In Opava, unusual cold periods from June 1815 continued up to December 1816. A shortage of grain resulted in a decree banning the distillation of spirits, issued on 13 November (Kreuzinger, 1862). Anton Lehmann reports imports of grain (with the exception of oats, which had a good yield) from Silesia, transported there from Russia where the yield, together with that of Poland, had been good (S6). This is also

confirmed by *Prager Zeitung* (6 October 1816, p. 1113) reporting transport of Russian grain to Trieste in Italy. However, a terse note from Hausner in his annual summary for 1816 reads: "*Hunger is inevitable.*" (S4).

**The year 1817**

According to Šimon Hausner, severe frosts occurred in Buchlovice between 8 and 16 January 1817; they followed on

from a thaw and were replaced by variable weather. Changeable weather with fewer frosts prevailed in February as well, when roads were muddy (S4). A flood on the River Vltava in Prague is reported for 7 March (Brázdil et al., 2005a). March is described by Hausner as an unpleasant month with daily frosts, snow and rain making roads muddy.



April 1817 was especially remarkable, described as a month with few fine days, continuous frosts, cold winds, incessant snowfall, very muddy roads and such awful weather that "*almost no previous April* [since 1803] *has been as bad*". After sleet on 7 April, Hausner reports 13 days upon which snow fell and a further three of precipitation – one with drizzle, one with rain and one with sleet, between 11th and 28th April (S4). Reports from Vienna are similar. Cold weather set

in on 11 April and snow fell almost daily between 17th and 28th April (*Wiener Zeitung*, 8 May 1817, p. 421). Kreybich, the Žitenice cleric, reports four landslides in spring, the result of extremely wet conditions in north-western Bohemia: the first on Křížová hora Mt. north of Žitenice, the second on Trojhora Hill between Chudoslavice and Třebušín, the third at Vitín near Malé Březno (community now defunct) and the fourth east of Jílové (S3). A fifth landslide is reported at Bohyně (east of Jílové) at the end of November, in addition to Kreybich (S3), by the *Prager Zeitung* from 22

December 1817 (p. 1403). May was recorded as too wet to work on the fields in Noviny pod Ralskem (S6).

      All the Czech documentary sources speak of shortages and rising prices in 1817. The high prices continued until the harvest of 1817, with shortages of food so severe that people milled rotting oats for flour (Trnka, 1912). A chronicle from Velká Bystřice reports that even when grain was available, there was insufficient money to buy it. It also records a far higher number of beggars than had been seen for many years (Roubic, 1987). The situation was

significantly ameliorated by a good harvest (a very high potato yield, for example, was reported for Boskovice – S5). However, Litoměřice had a below-mean grape vintage, in terms of both quality and quantity (Katzerowsky, 1895).

      The qualitatively-described increase in prices may be confirmed by actual records of prices for the basic grain crops. Data from Litoměřice in Bohemia (available only until 1817) and for Moravia, indicate bad harvests in 1815 and 1816 driving prices up from 1813 onwards, culminating in 1817. While in Moravia grain prices rose threefold (doubling

for oats), the figures for Litoměřice were fivefold for rye and barley and tripled for wheat and oats. Again the better harvest of 1817 drove prices down sharply, to the level of 1813 or below (Fig. 6). While prices for wheat, rye and barley exhibited similar steep increases and decreases, fluctuations in those for oats were more stable, also due to a good yield in 1816 (S6).

**4.2 Climatic, weather and social responses to the Lakagígar eruption**

The Lakagígar eruption in Iceland started on 8 June 1783, when a fissure opened up along the Laki crater-row and by February 1784, it had been followed by the largest terrestrial lava flow for the last millennium (Thordarson and Self, 1993, 2003; Stevenson et al., 2003). Moreover, Mt. Asama in Japan erupted at the beginning of August 1783 (Aramaki, 1956, 1957; Zielinski et al., 1994; Demarée and Mikami, 2005). Both eruptions have been classified as VEI-4.


**4.2.1 Climatic responses**

Fig. 7 shows seasonal temperature anomalies for Prague-Klementinum and central Europe series expressed with respect to the 5-year pre-Lakagígar eruption period. The eruption year was followed by a very cold winter in 1783/1784 and quite a cold spring in 1784. The cooling was further clearly expressed from winter 1784/1785 (particularly cold

January) to summer 1785 (with notably cold spring and March–April) and then from spring to autumn 1786 (particularly cold summer and autumn). Negative anomalies at Prague-Klementinum appeared for winter 1785/1786 and continued into spring 1787.

**4.2.2 Post-volcanic weather and impacts on society**

The Lakagígar eruption had clear responses in the presence of dry fog, unusual optical phenomena and heavy thunderstorms. The dry fog in central Europe was first observed on 16 June 1783 in Germany and western Bohemia (Stothers, 1996) as well as at Prague-Klementinum, where it occurred in 11 days of June, in 24 days of July and in 7



days in August 1783 (Strnadt, 1785). Visual daily weather observations kept by the reverend Karel Bernard Hein in Hodonice report fog from 27 June to 1 July and then in a further 12 days of July (Brázdil et al., 2003). Fog smelling of sulphur is reported from 18 June to 18 July, together with thunderstorms without rain, at Noviny pod Ralskem (Wiechowsky, 1928).

Jiří Vrbas from Písečné (Paměti starých písmáků moravských, 1916) also described optical phenomena in addition to dry fog: "*In 1783, every day and night from the festival of St. John* [24 June] *to the harvest, there were such dense fogs everywhere that nothing could be seen beyond a tiny piece of the world. The sun and the moon changed. The sun rose blood-red every day and the moon was like a black sack.*" Many reports also describe peculiar colours of the sun or moon at rising and setting. In Prague, a red sun (moon) is reported for 19 days from June to September (also four

days with aurora borealis) (Poznámky, 1977). A report from the chronicle of Antonín Kodytek from Kunvald (Nezbeda and Šůla, 1970) adds: "*… in summer there was such heat that had there not been unusual fog that shaded the sun, everything might have been burnt by the sun's heat.* […] *the rising sun could not be seen due to fires and then from six to nine o'clock* [it] *looked like a red-hot iron ball, then from nine to three or four o'clock it shone more intensely, but looked sad, which people held in awe.*"

Heavy thunderstorms, often without rain also attracted considerable attention. For example, Hein records 17 days with thunderstorms in summer 1783, the highest number during his observations in the 1780s (Brázdil et al., 2003). Systematic meteorological observations in Prague-Klementinum report thunderstorms on 23 days (Strnadt, 1785). Among the heavy thunderstorms of summer 1783, an event on 29 June in Klatovy stands out. For example, the chronicle of the Šebesta family (Hostaš, 1895) says: "[…] *there started such a downpour that it was if water were*

*poured out of a bucket* [and]*, at the same time there was such a thunder and storm uninterrupted from one o'clock in the afternoon until the evening and the lightning struck more than a hundred times.*" The lightning set fire to the armoury and its gunpowder. The resulting explosion killed and injured several people and damaged buildings in the surroundings. Six people were killed while ringing town bells, a practice commonly held at the time to avert thunderstorms, on 4 July in Doubrava and other human casualties were reported in Litoměřice, Terezín and Roudnice

nad Labem (for more details see Brázdil et al., 2003; Soukupová, 2013). These fatalities probably contributed to an imperial ban on ringing bells against thunderstorms, decreed on 26 November 1783 (Pán, 1931). That this measure was not fully observed is evident from records of money paid to bell-ringers in Hlinsko in summer 1786 (Adámek, 1917).

An example of the comprehensive nature of some of the descriptions of the weather effect after the volcanic event may be found in the chronicle of the Fuchs family from Třebětice (Verbík, 1982): "*For the entire summer of 1783*

*there were big, dense fogs with some* [kind of] *stench arising out of them, and thunder was heard but no clouds were visible. In the morning, during sunrise, the sun was as red as blood and this continued for a long time. And in the afternoon again, towards evening, it was the same. There was great drought in this year, constant thunder but no rain.*"

Significant flooding occurred in the post-Lakagígar years. Particularly notable was a disastrous flood in central Europe as February turned to March 1784, after the very severe and snowy winter of 1783/1784 (Brázdil et al., 2010a).

A flood on the River Vltava at Prague arising out of snowmelt and ice movement, culminating on 28 February 1784, was accompanied by loss of life and extensive damage. Floods were also recorded for many places on the Elbe and Ohře rivers in Bohemia. Ice movement also contributed to a Vltava flood on 16–17 April 1785, which lasted for ten days, following the coldest March in Prague temperature observations (from 1775 to the present). Another Vltava flood, on 17 August 1786 at Prague, was due to continuous rainy weather since 7 August; it left fields flooded and bridges

destroyed (Brázdil et al., 2003, 2005a). All three floods were also reported by documentary sources for the River Dyje in southern Moravia (Brázdil and Kirchner, 2007).





## 5 Discussion

### 5.1 Weather and climate effects of Lakagígar and Tambora eruptions

The climatic effects of the Lakagígar and Tambora eruptions may also be compared from the long-term viewpoint by ordering the mean seasonal temperatures in increasing series of seasonal values over the 1775–2015 period (Table 1).

Prague-Klementinum demonstrates the more extreme patterns post-Lakagígar: the coldest spring in 1785 and coldest summer in 1786, with the second-coldest autumn in 1786 and the third-coldest winter 1783/1784. Among the seasons following the Tambora eruption, only summer 1816 stands out, as the fifth-coldest (summer 1815 was 11th-coldest, tied with three other years). Central European temperature series averaging data over a broader area agree with Prague for spring 1785 and autumn 1786, but discloses summer 1816, post-Tambora, as absolutely the coldest in the 1775–2007

period. Furthermore, the general central European view significantly weakens the position of the cold summer of 1786 and enhances that of the cold summer of 1785 and spring 1817 compared with Prague. This reveals that spatial averaging of data and moving the territorial area of interest south-west of Prague (where the majority of the stations used for calculation of the central European series are located) may strengthen the summer signal of volcanic eruptions, including large tropical eruptions. While extremely cold springs, summers and autumns express clearly after both

eruptions, the two cold winters after Lakagígar have no analogue after the Tambora event.

      In terms of individual months, those following the Lakagígar eruption appear among the ten coldest years seven times in the two series and those following Tambora four times. The coldest months to appear in both complete series were March 1785 and April 1816; the coldest at Prague-Klementinum alone was July 1786. The second-coldest in the Prague series appear in January 1784 and August 1786, while this position is occupied in the central European

series by April 1785 (in Prague series, April 1785 is third) and October 1784. Among the ten coldest years were also July 1815 (Prague-Klementinum only), July 1816 (central Europe only), August 1816 and October 1817 (both series). A composite of monthly temperature anomalies for both volcanic events appears in Fig. 8.

      In Büntgen et al. (2015), the summers of the 1810s constitute the coolest decade in central Europe in the past three centuries, based on the analysis of tree-rings in Swiss stone pine. Cole-Dai et al. (2009) refer to this time as

probably the coldest decade in the last 500 years or more in the Northern Hemisphere and the tropics. However, these findings are not confirmed by the temperature series used in this study. In central European temperature series based on documentary and instrumental records (Dobrovolný et al., 2010), the 1810s summers were third-coldest after the 1690s and 1910s (in the 1500–2007 period). In the Czech Lands, the series from Brno was fourth-coldest (1800–2010) and those from Prague-Klementinum and mean Czech areal series the fifth-coldest (1780–2010 and 1800–2010

respectively).

### 5.2 Explanations of post-volcanic weather and climatic effects

The clearly-observable and unusual weather effects such as dry fog, optical phenomena and heavy thunderstorms without rain after the Lakagígar eruption stimulated a large body of discussion. For example, the chronicler Johann

Josef Langer from Rýmařov attributed them to an earthquake in Messina (Italy) on 5 February 1783 (Tutsch, 1914): "*During the following summer* [1783]*, outstanding fogs were observed in the European countries, here one time, there the next, also in the surroundings. The cause of the fog was attributed to this* [Messina] *earthquake.*" News of the eruption, brought by vessels of the Danish trading monopoly from Iceland to Copenhagen, did not reach Europe until 1 September. Benjamin Franklin (1706–1790), American diplomat, scientist, inventor and writer, tried as early as May

1784 to explain the dry fog observed in the summer months of 1783 and the severe winter of 1783/1784 as follows: "... *or whether it was the vast quantity of smoke, long continuing to issue during the summer from Hecla, in Iceland, and that other volcano which arose out of the sea near that island, which smoke might be spread by various winds over the*





*northern part of the world, is yet to ascertain*" (Humphreys, 1913). A year later he was probably the first to connect climate with volcanic eruptions (Demarée and Ogilvie, 2001). Observations on the Asama eruption were published by the Dutchman Isaac Titsingh in Dejima (Nagasaki) in the early 19th century (Demarée et al., 2013).

5        Reactions to the Tambora eruption were different. Czech documentary sources recorded no remarkable weather phenomena. However, the series of cold summers from 1812 onwards, and particularly that of 1816, led to speculation about possible causes. The newspaper *Wiener Zeitung* of 9 July 1816 (p. 755) and the *Brünner Zeitung* of 12 July 1816 (pp. 759–760) reprinted an article from a certain Böckmann from *Badischen Staatszeitung*, responding to the series of consecutive cold summers after the warm summer of 1811. First he mentioned Flaugergues' comet (Fig. 9a) observed in the autumn of 1811: "… *the large, remarkable comet of 1811 had a particular influence on our solar*

*system, and* […] *stimulated physical processes in the Earth's atmosphere, hitherto unknown, through which unusual warmth was generated, concurrently perhaps leaving* [certain] *substances or removing others, which otherwise in usual circumstances, particularly in summer weather, might not have had such a visible influence.*" His article also discusses the possibility that the cool summers may have resulted from the number of sunspots (Fig. 9b): "*Therefore certain natural scientists believed an explanation of the cold years* [might be found in that] *during them the Sun produced less*

[sun]*light and in warm* [years] *more than usual.* [...] *These views themselves are, however, not yet proven; we have had hot summers with many sunspots and cold winters with few.*" A slow cooling of the Earth is mentioned as a third possible trigger: "*From another side, an explanation for the unusual cold weather has been sought for many years in the fact that Earth was once very hot and should now be getting cooler. Were such cooling alone real, so our mean annual warmth would be reduced by only one degree for every 10,000 years that have passed, and thereby our climate*

*would be similar* [in the same steps]*, over such a large time interval, to the climate which is* [now] *in the area situated about 70 hours to the north.*"

        However, remarks made by the I. R. Astronomical Observatory [*k. k. Sternwarte*] in Vienna concerning the extremely cold and snowy second half of April 1817, published in *Wiener Zeitung* on 8 May 1817 (p. 421), turn attention to earthquakes and volcanic eruptions rather than to comets or sunspots: "*But earthquakes and volcanic*

*eruptions could well give rise to the origin of the recent* [snow and heavy thunderstorms in places]*, as a public newspaper has already mentioned. That ongoing chemical processes in the Earth interior and on its surface as well, due to changes in the atmosphere, give rise to various phenomenon that may be based on them, might be something more than mere surmise, and more probable than that comets, at distances of millions of miles, having in any event no case significant mass, as well as more or fewer sunspots, could have the ascribed effect on the weather patterns that are*

*appearing on our Earth.*" Finally, some contemporary scientists attributed the cold summer of 1816 in western Europe to huge masses of ice drifting in the North Atlantic (Bodenmann et al., 2011).

        The effect of the Tambora eruption on air temperature was mentioned marginally by Humphreys (1913) in a discussion of the role of volcanic dust and other factors in climatic changes. First he attributed the cold years of 1783–1785 to the explosion of the Japanese volcano Asama in 1783 and then mentioned that "*the "year without a summer,"*

*that was cold the world over, followed the eruption of Tomboro, which was so violent that 56,000 people were killed and "for three days there was darkness at a distance of 300 miles*".

### 5.3 Social impacts of the Lakagígar and Tambora eruptions
The ways in which the Lakagígar and Tambora eruptions impacted on society must be addressed in the light of the

contemporary socio-political situation. Emperor Joseph II, considered by many a representative of enlightened absolutism, ruled the Austrian Empire in the 1780s. His reign was notable for new freedoms – the abolition of serfdom, the beginnings of religious tolerance and the restriction of censorship. It was also a time of German cultural colonisation



("Germanization"), burgeoning bureaucracy and state centralization. The Czech Lands saw something of a national renaissance, cultural development (new schools, theatres, libraries, etc.) and economic advancement (establishment of the first factories, construction of imperial roads, etc.) (Bělina et al., 2001). Much of society in the Czech Lands was agricultural, centring on grain production; however, innovations to backward agricultural practices were slow to penetrate (Beranová and Kubačák, 2010). By end of this period, there was growing dissatisfaction throughout society, largely fuelled by rising prices and despotic approaches to the implementation of change (Lněničková, 1999).

Emperor Franz I, deeply conservative, was ruler of the Austrian Empire during the 1810s. He expanded royal power to penetrate every corner of society, creating what was essentially a police state, with rampant bureaucracy, censorship and resistance to reform (Taylor, 1998). The Czech Lands were the first part of the Austrian Empire to participate in the industrial revolution. Craftsmanship and manufacturing gathered pace, and agriculture took to the rotation of crops. The Napoleonic wars marred the first five years of the 1810s, accompanied by stagnation of population growth, rising prices, poverty, hunger, increasing numbers of beggars and higher incidence of unrest in the countryside. Constant warfare led to state bankruptcy in 1811 (Bělina et al., 2013). However, change was not far off. Demand for grain and foodstuffs rose and with it prices, leading to higher incomes for farmers, characteristic of the period. The internal situation calmed down after 1815, demand for foodstuffs increased still further, agriculture developed and population growth revived. There was an agricultural boom in the Czech Lands that lasted until 1817 (Lněničková, 1999). Albert (1964), investigating an agricultural crisis after 1817 in Moravia, explains the increase in grain prices after the Napoleonic wars in terms of agriculture intensification, He posits that expanding potato cultivation started to compete with grain, and livestock numbers were low, insufficient to absorb any corn-growing surplus. He did not associate the less productive years with the Tambora eruption. A drop in grain prices after 1817 in Moravia was related to good harvests in 1818–1821. Farmers' incomes fell in response to decreasing prices and they found themselves unable to meet taxation demands.

On the other hand, Post (1970) attributed the growth in grain prices that followed the Napoleonic wars in Europe to, apart from inflation and overproduction, the barren years of 1816–1817 resulting from low temperatures and abundant precipitation related to volcanic eruptions, particularly of Tambora. The subsequent drop in prices led to a series of bankruptcies, poverty and vagrancy; this situation was reflected in population decline and increased mortality.

Differences between the Lakagígar and Tambora post-volcanic periods clearly follow from comparison of the prices of key cereals. A composite of prices for Litoměřice in north-western Bohemia (Fig. 10a) and Moravia (Fig. 10b) demonstrates a clear increase in prices in the post-Tambora years up to 1817 with a drop thereafter, while no such trends emerge in the post-Lakagígar years: the prices of barley and oats even increased in Litoměřice, while wheat and rye cost more in line with inflation. Brno, one of the largest cities in Moravia, recorded a significant increase in cereal prices in the late 1780s. Aside from inflation, this could be attributed to the compulsory supply of grain to military stores, used to feed the military in transit and on long-term station, leading to grain shortages (Petráň, 1977). According to František Jan Vavák, such movements took place in Bohemia in the years 1786 and 1788 in particular (Skopec, 1910, 1912).

## 6 Conclusions

The literature addressing the climatological and environmental consequences of large volcanic eruptions at various spatial and temporal scales is extensive. The eruption of Tambora in April 1815, the strongest, at VEI-7, has attracted the most attention and widespread interest in its impacts, particularly in 1816, the "Year Without a Summer". The analyses and documentation cited in this paper demonstrate relatively weaker effects at regional or local scales for central Europe (e.g. Briffa and Jones, 1992; Písek and Brázdil, 2006) compared with certain closer but less intense




eruptions, such as the Lakagígar eruption in Iceland in 1783–1784. This has also been confirmed by Mikšovský et al. (2014), who revealed the prominent and statistically significant imprint of major volcanic events on the global temperature signal while changes in mean Czech temperature series remained negligible (1866–2010 period).

The main features of the two eruptions and their impacts on the Czech Lands are summarised in Table 2, which makes clear that the effects of the Lakagígar eruption were climatologically stronger than those of the Tambora eruption, while the opposite held for societal responses. The Tambora eruption was far harder on society, to the point of famine in some central European countries such as Germany (Bayer, 1966) and Switzerland (Krämer, 2015). Post (1977) even spoke of this time as "the last great subsistence crisis in the Western world". However, the impacts on life in the Czech Lands in the post-Tambora years were not comparable with the "Hungry Years" of 1770–1772 (Brázdil et

al., 2001; Pfister and Brázdil, 2006) or with other known, massive famines before AD 1500 (in the 1280s, 1310s and 1430s – see Brázdil et al., 2015).

**Acknowledgements**

The authors acknowledge the financial support of the Grant Agency of the Czech Republic for project no. 13-19831S.

RB, LŘ and LD received funding from the Ministry of Education, Youth and Sports of CR within the National Sustainability Program I (NPU I). Stefan Brönnimann (Zürich) and particularly Gaston R. Demarée (Brussels) are acknowledged for recommending references, Pavel Raška (Ústí nad Labem) for help with localisation of landslides in north-western Bohemia, and Tony Long (Svinošice) for helping work up the English.

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

(S5) Státní okresní archiv Blansko, fond Archiv města Boskovice, inv. č. 109: Kronika Dominika Kučery.

(S6) Státní okresní archiv Česká Lípa, fond Sbírka rukopisů, sign. 13/3: Witterungs-Geschichte. Auszug aus den Titl: Lesenwürdige Sammlungen der hinterlegten Jahrgängen. Von Anton Lehmann Lehrer in Neuland. Abgeschrieben durch Joseph Meißner.



**Table 1: The order of mean seasonal temperatures at Prague-Klementinum and in central Europe series, in order of increasing values, for post-eruption years after Lakagígar (1783–1786) and Tambora (1815–1818). The duration of the full temperature series is 241 years for Prague (240 for winter) and 233 years for central Europe.**

| Year | Prague-Klementinum | | | | Central Europe | | | |
|---|---|---|---|---|---|---|---|---|
| | Winter | Spring | Summer | Autumn | Winter | Spring | Summer | Autumn |
| Lakagígar eruption | | | | | | | | |
| 1783 | - | 99–104 | 147–157 | 132–146 | - | 123–128 | 164–173 | 130–137 |
| 1784 | 3 | 39–44 | 74–81 | 125–128 | 6–8 | 59–64 | 98–100 | 61–67 |
| 1785 | 22–23 | 1 | 20–27 | 117–124 | 24–25 | 1 | 9–12 | 124–129 |
| 1786 | 59–60 | 24–30 | 1 | 2 | 100–104 | 65–70 | 9–12 | 2 |
| Tambora eruption | | | | | | | | |
| 1815 | - | 189–195 | 11–14 | 36–45 | - | 198–201 | 9–12 | 36–40 |
| 1816 | 90–91 | 92–98 | 5 | 21–26 | 60–63 | 46–58 | 1 | 29–32 |
| 1817 | 185–188 | 16–18 | 93–107 | 71–79 | 183–191 | 8–9 | 73–84 | 89–97 |
| 1818 | 149–154 | 174–177 | 66–73 | 125–128 | 142–148 | 149–156 | 73–84 | 107–116 |

**Table 2: Summary comparison of the consequences of the Tambora (1815) and Lakagígar (1783) volcanic eruptions in the Czech Lands.**

10   Tambora 1815                                           Lakagígar 1783

| Temperature: One extreme summer, 1816 | Temperature: A number of extreme seasons in 1784–1786 |
|---|---|
| Precipitation: Several wet seasons | Precipitation: No measured data |
| No directly observed post-volcanic weather effects | Dry fog, heavy thunderstorms (no rain), sun and moon red |
| Bad grain harvest, rise in grain prices | No effects on agriculture and grain prices |
| Lack of bread, hunger, high vagrancy | No crisis indications |
| Floods: August 1815, March 1817 | Floods: February 1784, April 1785, August 1786 |
| Landslides in north-western Bohemia (1817) | No indications of landslides |
| Important natural and societal impacts | Fatalities among those ringing city bells to ward off thunderstorms – prohibition of ringing |



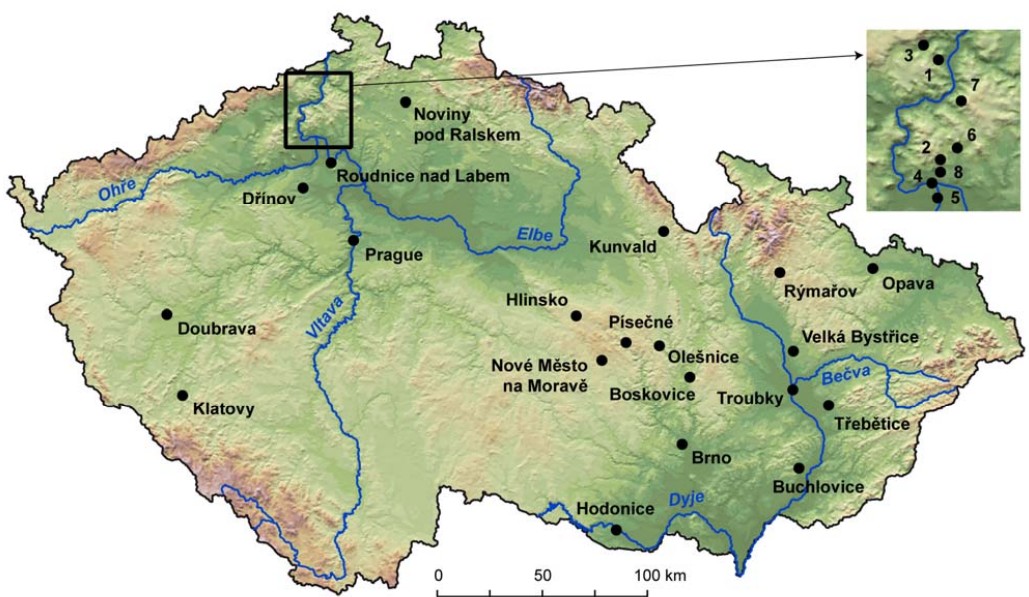

**Figure 1: The Czech Republic, showing places from which meteorological observations and documentary weather reports were obtained for the purposes of this study: 1 – Bohyně, 2 – Chudoslavice, 3 – Jílové, 4 – Litoměřice, 5 – Terezín, 6 – Třebušín, 7 – Vitín, 8 – Žitenice.**



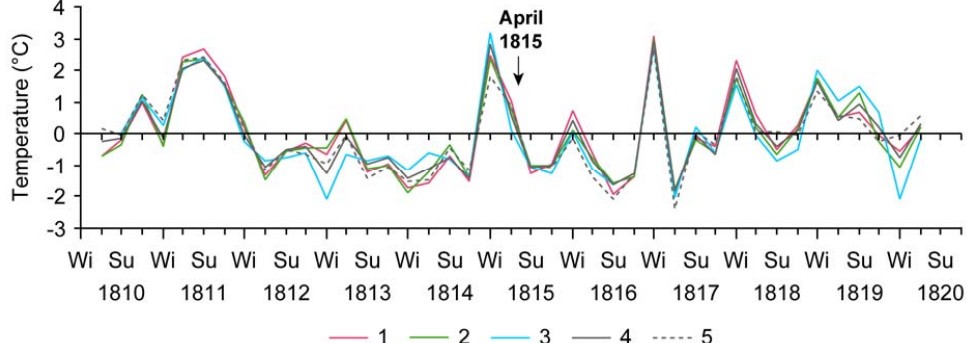

**Figure 2: Seasonal temperature anomalies at Prague-Klementinum (1), Žitenice (2), Brno (3), in the Czech Lands (4) and central Europe (5) series around the time of Tambora eruption in 1815 (Wi – Winter, Su – Summer). Anomalies are expressed with respect to the 5-year period pre-eruption: month zero – April 1815.**




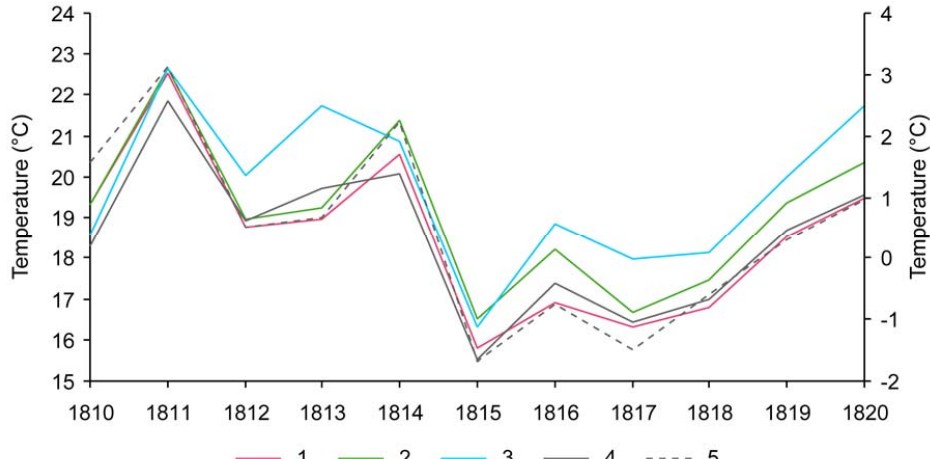

**Figure 3: Differences in mean summer and winter temperatures in Prague-Klementinum (1), Žitenice (2), Brno (3), Czech Lands (4) and central Europe (5) series, demonstrating reduction of seasonality after the Tambora eruption in 1815. Central European differences are calculated from temperature anomalies with respect to the 1961–1990 reference period (right axis).**





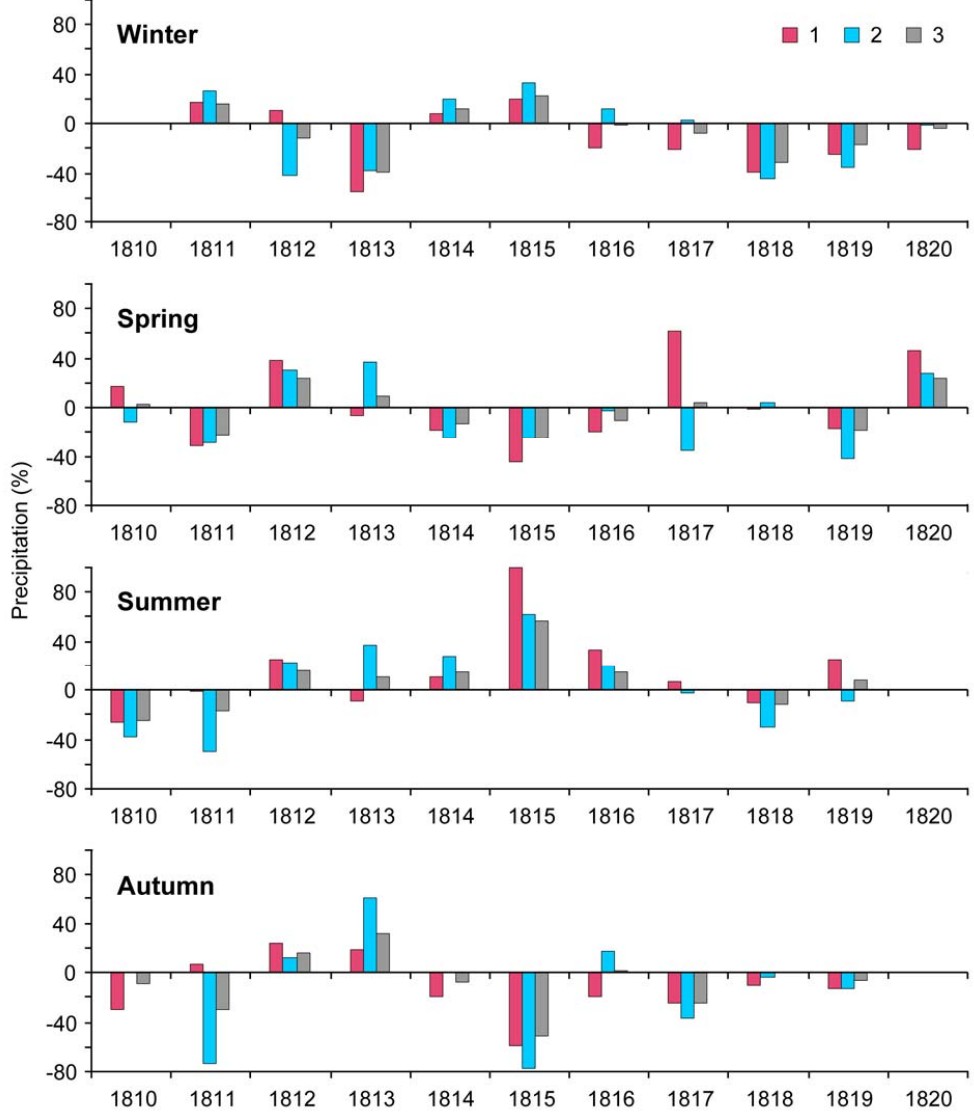

Figure 4: Seasonal precipitation anomalies in Prague-Klementinum (1), Brno (2) and Czech Lands (3) series around the time of Tambora eruption in 1815. Anomalies are expressed with respect to the 5-year period pre-eruption: month zero – April 1815.





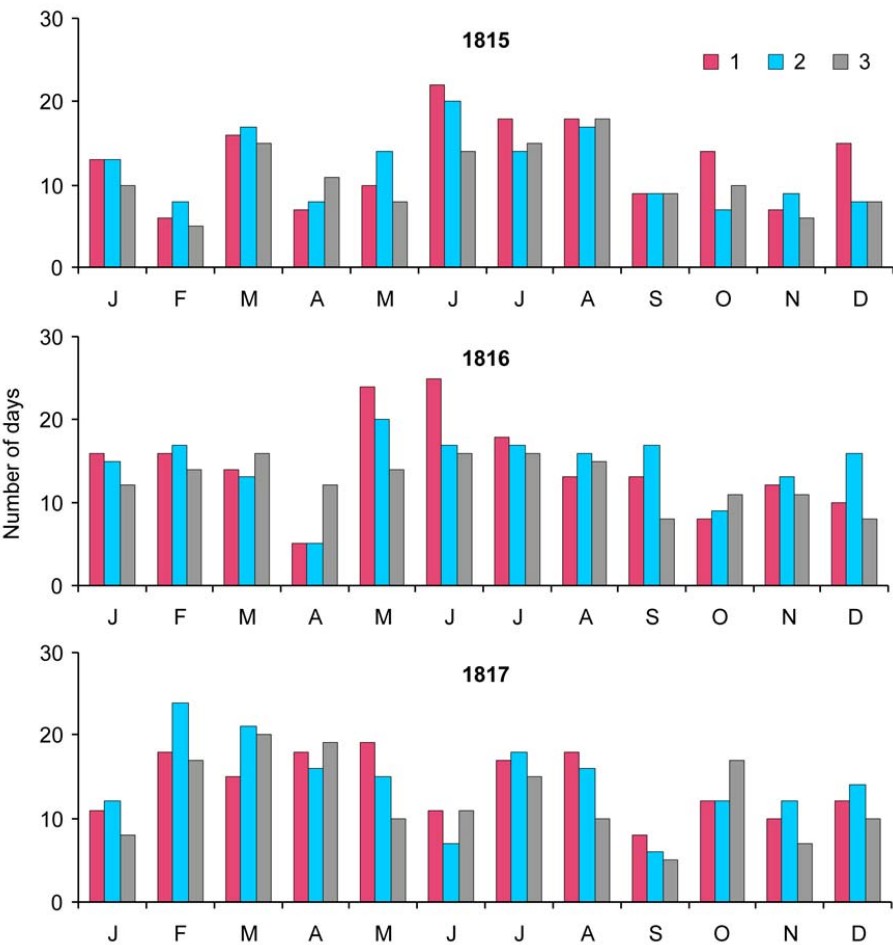

**Figure 5: Fluctuations in the monthly numbers of precipitation days in 1815–1817: 1 – Prague-Klementinum (Meteorologická pozorování, 1976), 2 – Žitenice (S1–S3), 3 – Buchlovice (S4).**





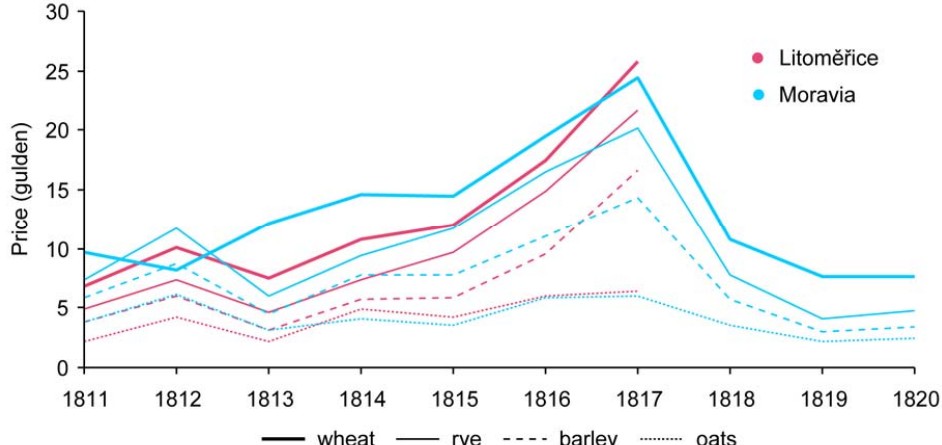

Figure 6: Fluctuations in prices of four staple cereals (wheat, rye, barley, oats) in Litoměřice, north-western Bohemia (gulden/61.49 l) and in Moravia (gulden/hl) in the 1811–1820 period. Data: Litoměřice – Tlapák (1977) (only to 1817), Moravia – Albert (1964).



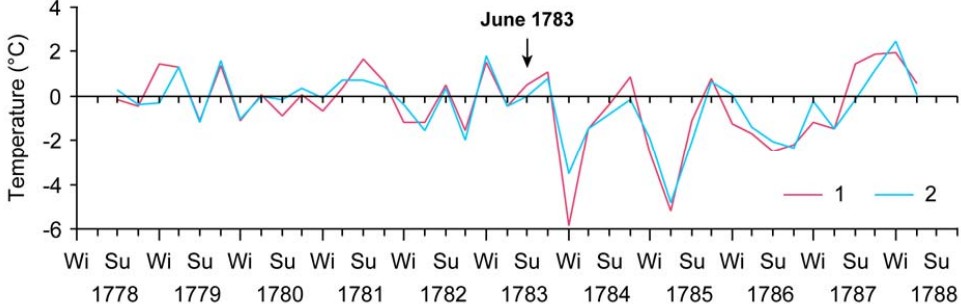

**Figure 7: Seasonal temperature anomalies for Prague-Klementinum (1) and central Europe (2) series around the time of the Lakagígar eruption in 1783 (Wi – Winter, Su – Summer). Anomalies are expressed with respect to the five years pre-eruption: month zero – June 1783.**





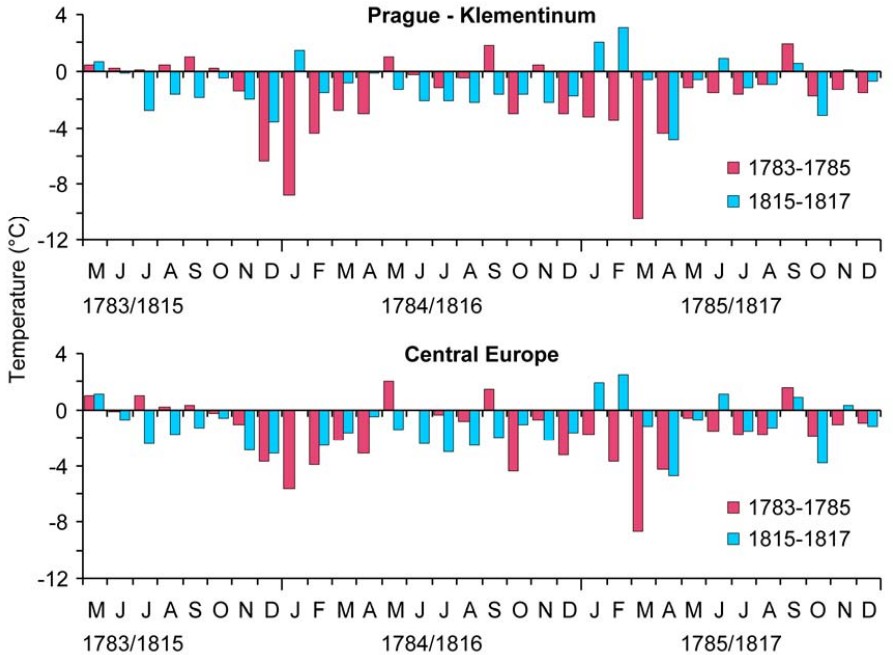

**Figure 8: Composite of monthly temperature anomalies (with respect to the 1961–1990 reference period) in Prague-Klementinum and central Europe series in 1783–1785 and 1815–1817.**



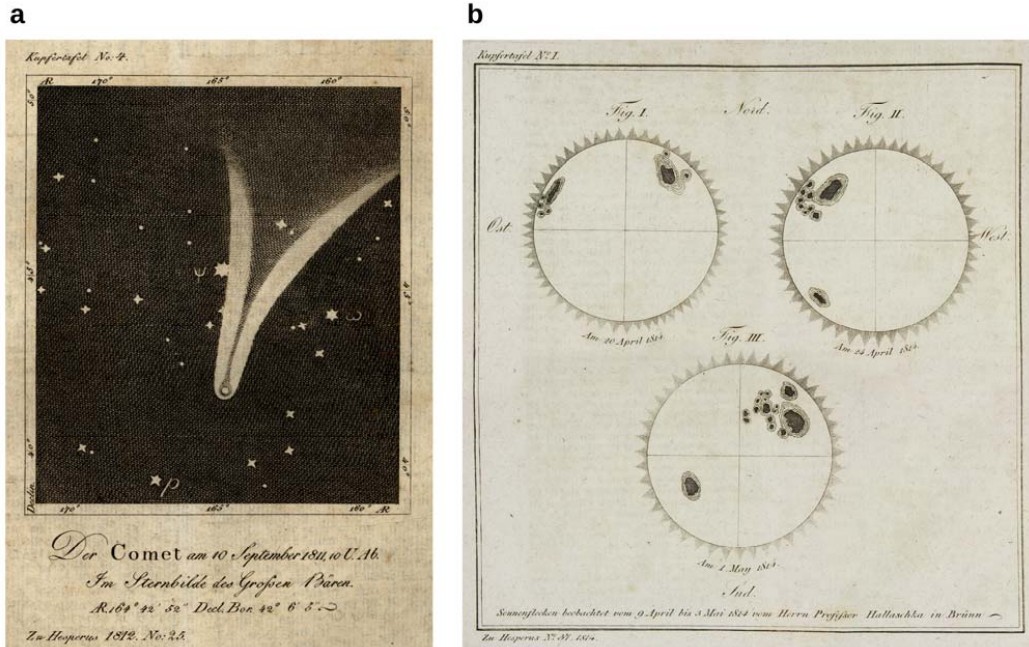

**Figure 9: a) Flaugergues' comet observed in September 1811 was, according to many people, responsible for the dry weather and a portent of its continuation. The illustration shows the position of the comet on 11 September at 10 p.m. in projection to the constellation of Ursa Major, recorded in Brno (attachment to *Hesperus*, 1812, no. 25); b) Sunspots observed by Cassian Hallaschka in Brno from 9 April to 3 May 1814 (supplement to Hallaschka, 1814).**





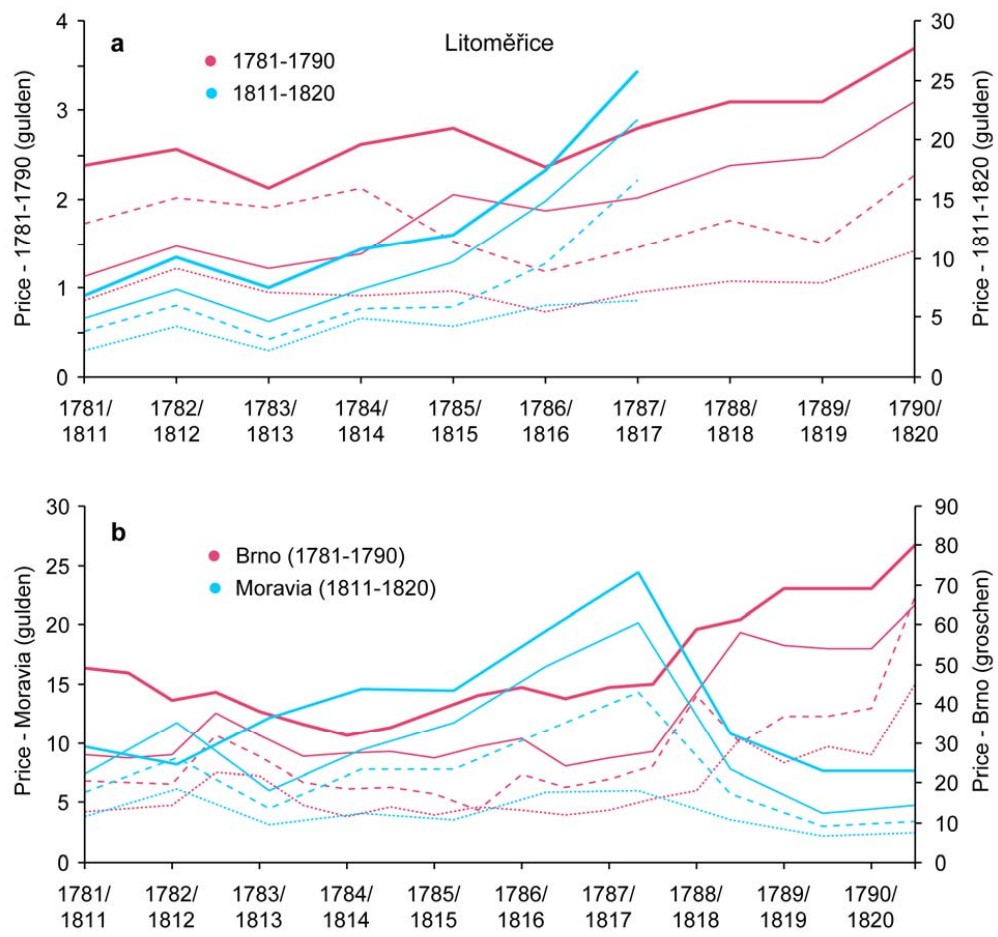

**Figure 10: Composite of fluctuations in prices of the four main cereals (wheat, rye, barley, oats) in the 1781–1790 and 1811–1820 periods: a) Litoměřice (gulden/ 61.49 l) – data only up to 1817 from Tlapák (1977); b) Brno (groschen) – data from Brázdil and Durďáková (2000), Moravia (gulden/hl) – data from Albert (1964).**