# Peer review of "Climatic effects and impacts of the 1815 eruption of Mount Tambora in the Czech Lands"

_Climate of the Past, 2016_

## Referee Comment (RC1) · Anonymous Referee #1 · 7 Mar 2016

This paper compiles the effects of the Tambora and Lakagígar eruptions on the Czech Lands. The authors have a great trajectory in the use of documentary sources with climatic objectives, mainly in Czech Lands. For this reason, some of the information presented has been previously published in different works with different objectives. The paper is a little bit confusing and it is not publishable in its current form.

Specific comments

The authors should reconsider the importance/necessity of the comparison with the Lakagígar eruption. Both eruptions are completely different (latitude, date, vicinity to Czech lands. . .) so the authors should clearly explain why it is interesting this comparison between them. Moreover, it is important that the authors explain clearly the

different features of the two eruptions. In this point, I think that another option is focusing the paper only in the Tambora eruption.

The discussion section is not clearly linked with the result sections; this is more evident in sections 5.2 and 5.3

One of the main conclusions of the paper is that the Tambora eruption impacted less in the climate and more in the society that the Lakagígar one. But I miss a discussion about why this happened.

Introduction The unidentified eruption of 1809 is cited in the introduction. But nothing about this eruption is explained in the rest of the text. This eruption can affect the short-term analysis presented in the paper because "the mean temperature for each month was calculated using temperature data from five years prior to the eruption", some discussion about that could be interesting.

About the impact of Lakagígar out of Europe could be interesting to cite Trigo et al (2010). Also could be useful in the discussion about the foggy events. Ordering the archival sources the S1 must be cited the first in the text then S2. . .

Methods No methods are described for the use of the documentary data (no instrumental).

Results Pag. 3 line 33-37 This paragraph would be better in the introduction with a comparison with the Lakagígar eruption. I do not like the structure. I think that some information given in "Post-volcanic weather and impacts on society" are "climatic responses". I propose a year by year structure but with all the information (instrumental and documentary, climatic and social) for each year. Pag 4. Line 29. When are the haymaking and the grain harvest? Page 4 line 29-30 "if two days were fine, it then rained for two days." This phrase it is not clear for me, is it referred to august?. Page 4 line 30 "The wine vintage was bad for the third year" I do not understand this phrase, what year is the third? 1815? Is there some climatic explanation for the caterpillars

plague in May? Pag. 4 line 36-37. "Kreybich reports a flood on the Elbe for 10–14 August with extensive damage to agricultural crops" is it known the specific location? Zitenice? Pag 4. Line 41. The dry autumn of 1815 is also clear identified in figure 4. Pag 5 line 11 "Other Czech documentary sources report 1816 as particularly cold and wet, with bad harvests and rising prices of all products" this phrase need a cite. Pag 6 line 11 "shortages" of what? food? water? Pag. 6 25-29. I see better this paragraph in the introduction and developing a comparison with the Lakagígar eruption. Pag. 7 Many references to thunderstorms during the Lakagígar eruption but also during the Tambora. Can you discuss deeply how this phenomenon could be induced by the eruptions?.

Figures Figure 1: It would be interesting including a legend to explain which locations have instrumental information (temperature and precipitation) and/or documentary information. Figure 2, 3, 4: Does it make sense including the Chez Lands series? This series during this period is calculated from Prague and Brno. Both included in the figures. Figure 6: Redundant, all the information in this figure is also in figure 10. Technical comments Be coherent with format of the dates 7 April or 28th April.

Trigo R.M., Vaquero J.M., Stothers R.B. (2010). Witnessing the impact of the 1783-1784 Laki eruption in the Southern Hemisphere. Climatic Change, 99, 535-546.

──────────────────────

---

## Referee Comment (RC2) · R. Trigo (Referee) · 21 Mar 2016

**Effects on the Czech Lands of the 1815 eruption of Mount Tambora: responses, impacts and comparison with the Lakagígar eruption of 1783**

The paper provides a review on the impacts in Czech Lands of two major eruptions with well-known impacts in European climate; Tambora (1815) and Lakagigar (1783). The authors use a wide variety of datasets (observations and documentary) to put into a wider and long-term perspective analysis. Although some parts of the manuscript are interesting, the overall structure is not too clear and novelty and key results are not particularly thrilling.   Thus, I consider that the paper should not be accepted as it is and should be reconsidered after the authors improve it substantially.

**1.  Major comments**

**1.1. (Novelty of datasets used and results obtained)** It is not entirely clear to readers the level of novelty of the various datasets presented in section 2.2. If I understood correctly all datasets have been described/used in the past, with the exceptions of the documents related to Reverend Simon Hausner and the teacher Noviny pod Ralskem. This is important to understand if the authors have simply used datasets compiled previously (even if often by themselves) or if new datasets where explored within the scope of this particular work. **Please clarify.**

Moreover, to ensure reproducibility and homogenization of derived datasets it is common for authors to provide all methodological steps on the information and time series derived from documentary sources. Here no such information is provided in section 3 (Methods), underlining perhaps that these are not new datasets (?). **Please clarify.**

It is clear that the authors have a large experience in past-climate analysis, particularly over Czech Republic. Thus, it is expected that all relevant literature for the main topic of this work (i.e. impacts of major eruptions in Czech lands) is provided at the introduction, allowing to stress the novelties that will be investigated here. Thus it is rather strange that the first time a key reference evaluating the impact of major eruptions in the mean Czech temperature region is mentioned only at the end (Page 11), and not in the introduction (Mikšovský et al, 2014). **Please clarify.**

**1.2. (Lack of statistical significance inference of several results).** There are a number of interesting results describing weather/climate extremes that may be associated to the effects of both eruptions in the climate of the Czech Lands. However, many times the descriptions are not accompanied by a more robust statement on the statistical significance (or uniqueness) of the so called-extreme event. A few examples of that are highlighted here:

a) (Page 4, lines 33-36): "A message from Litoměřice dated 9 August reports a flood lasting eight days on the River Elbe after five weeks of rainy periods. The water rose to a level of two feet [c. 65 cm] under the bridge, so the structure survived, but grain, vegetable and other field crops were damaged (Katzerowsky, 1895)." **How exceptional is this situation? How many time has it occurred in the last 300 years?**

b) (Page 5, lines 8-10): "The ice was definitely gone by 8–9 March (S2). Lehmann reports a 3/4-ell [c. 58-cm]-thick crust of ice on some fields in Noviny pod Ralskem (S6). Frosty weather prevailed in March with blizzards from 26 to 31 March. April was cold and dry, with no heavy rain (S4).." **Again, to what extent are these descriptions unique in the longer term context?**

c) (Page 6, lines 5-8): Kreybich, the Žitenice cleric, reports four landslides in spring, the result of extremely wet conditions in north-western Bohemia: the first on Křížová hora Mt. north of Žitenice, the second on Trojhora Hill between Chudoslavice and Třebušín, the third at Vitín near Malé Březno (community now defunct) and the fourth east of Jílové (S3). **Are landslides very rare in the area? How often do these occur?**

**1.3. (The choice of Tambora vs Lakagigar is not clear).** It is not clear to readers the choice of these two eruptions that are so different in their characteristics, location, impacts, etc. A more straightforward approach would be to consider several major tropical explosive eruptions (as those listed in Fischer et al. 2007) or, alternatively, major eruptions in high latitudes (particularly in Iceland). Besides taking place roughly with 30 years apart, it is not entirely clear the rationale for the combined assessment. **Please clarify.**

Please notice that the differences between the two types of eruptions are so large that they have implications in the literature cited (that can be quite different) and even way their impact is evaluated. In particular the definition of month 0 (and in fact year

0, 1 and 2) is quite unclear to me in the case of the eruption of Lakagigar that took place between (1783 and early 1784). **Please clarify**

**2. Minor suggestions/comments**

**2.1.** (Page 3, sections methods) Please provide 1 or 2 references to support the various options explained, particularly the 5+5 years used before and after the eruption.

**2.2.** (Page 3, end of section 4.1) I think that this section would gain with a sentence explaining that major tropical eruptions (e.g. Tambora-1815, Krakatoa-1883, Pinatubo 1991) have the capacity to alter the radiative balance for the entire world, impinging widespread cooling at the surface level of the globe, but often inducing large-scale changes in the atmospheric circulation that can warm the continental areas in winter (see carefully Robock 2002, Science).

**2.3.** (Page 3, sections 4.1) The term VEI has not been described before. Please provide its meaning here when it appears for the first time (Volcanic Explosivity Index, VEI). It would be also useful to give a range of its scale between 1 and 8 (and a glimpse of the logarithmic nature of its scale, thus emphasizing the much larger volume of lava associated to a VEI-7 when compared to a VEI-6).

**2.4.** (Page 4, lines 1-2, Fig. 2) The 5 lines used in Fig.2 are very similar and it is not clear the exception mentioned for Brno as being particularly milder than the others for the winter 1816/1817 (?)

**2.5.** (Page 4, Section "The year 1815") Are the author implying that the "cold May 1815 with more frequent rain and frosts on 29–30 May" are related to the Tambora eruption? And the same doubt applies to the reference to the fruit trees eaten by caterpillar.

**2.6.** (Page 4, Section "The year 1815") Are the authors implying that the "cold May 1815 with more frequent rain and frosts on 29–30 May" are related to the Tambora eruption?

**2.7.** (Page 6, line 17) Please provide a reference to Fig.6 earlier at the end of the sentence: "…driving prices up from 1813 onwards, culminating in 1817 (Fig. 6)".

**2.8.** (Page 7, lines 33-38) Several specific extreme weather events are mentioned here (e.g. March 1784; April 1785). A number of works for other sectors of Europe have

been developed for the years post-Lakagigar, please provide some links to these works in terms of compatibility (or not) of the atmospheric circulation anomalies.

**2.9.** (Page 8, section 5.2) The contents of this section are not particularly well incorporated into the overall flow of the text. First, this discussion is not structured with Tambora being analysed after Lakagigar (that should be probably the most natural order, but the authors have preferred the reverse from the beginning). Secondly the temporal and spatial link between these various theories (earthquake in Messina 1783, Comet in 1811, Number of sunspots in 1814, etc) is not provided in a meaningful way.

**Figures**

Fig1 Please provide different symbols for stations with different information. For example Prague and Brno should have a distinct symbol. The same apply for those locations with just documentary sources.

Fig3. I believe the figure caption should read: "Difference between mean summer and winter…"

Fig. 6 It seems that the contents of this figure is repeated in Fig 10 (?)

Fig. 7I believe that the time delimitation of Lakagigar eruption should extend until February 1784.

**Ricardo Trigo**

---

## Referee Comment (RC3) · Anonymous Referee #3 · 22 Mar 2016

The main aim of this paper is to present the consequences of the 1815 Mount Tambora eruption in the Czech Lands and to compare it with the 1783 Lakagigar eruption. The authors use two kinds of data: i) instrumental data - temperature and precipitation long series of Prague-Klementinum and Brno, as well as others series; ii) documentary data including contemporary press referring to the extremely cold and wet summers of 1815 and 1816 (the "year without summer") and to grain production and prices. It is an important and informative paper. It also displays a synthesis of the effects of the Tambora and the Lakagigar eruptions in the Czech Lands. However, I think that the paper should be improved as follows:

1. There might be some problem in this paper's structure and contents. The authors

have centered their paper on the 1815 eruption and its consequences. But they also include information about the Lakagigar eruption mainly within the "results" section (only a short paragraph in the introduction). This is quite confusing. Even the title is too long and quite ambiguous. Moreover the two eruptions are rather different and so are their impacts.

I think the authors could consider two solutions

- A) Either concentrate on the 1815 eruption, as they possess more instrumental and documentary information

- B) Or write a paper on the comparison of the two eruptions and their consequences, change the title and modify the paper's structure accordingly, using and developing the texts where this comparison is already carried out.

2. Introduction - Explain how an eruption in tropical site may affect the climate in central Europe. If you include the L. eruption, compare the features of both eruptions.

3. Data section – Documentary data and the notes that accompany some of the instrumental data should be described in more detail. In some cases, the authors refer to their own past publications, but a short sentence could clarify the content of each of the sources (e.g. 1), p. 3, l.11). Indicate clearly the new information brought about by this paper.

4. The methods section (p. 3) should be more clear and developed, particularly when it comes to documentary data (different steps that were necessary to construct a dataset from the documentary data). This is included in other papers from the same authors but should be incorporated here referring to these specific cases.

5. The results sections (p. 3- 7) should be reorganised according to your choice of A) or B) (see above, please). Should not the comparison of the two eruptions referring to climate and to their impacts be included in the results part? (if you follow B. If you select A, then these paragraphs should be deleted).

6. Rewrite the discussion part adding some current explanations about the differences of the two eruptions and why are the impacts different (if you choose B), putting the events into European context.

p. 3, line 9 –Explain what are visual weather records

p.4, 2nd paragraph. As the authors notice there had been already a cool period in 1812-1814. Perhaps the authors should point out more clearly the differences between these two cold periods and the drop of temperature anomaly after 1815.

P. 4, line 16 (and Fig.4) – you refer that autumn 1817 shows strong negative anomalies, but autumn 1815 has also little rain. Please explain.

Figure 1 – indicate through different symbols the places from where you used meteorological and documentary data. If you have both data for the same site use a combined symbol.

Figure 2 – why are the anomalies calculated relatively to the five years' period pre-eruption?

Figure 3- The Figure caption is not clear and the two "temperature °C" in the vertical axes are confusing. You could write in the right one "Temperature anomalies in C.E." and leave the left one as it is.

Figures 6 and 10 – there is no need to include both figures.

---

## Author Response (AR1)

**RESPONSES TO REFEREES**
(responses to referees and and changes in the manuscript are in red colour)

We would like to thank **Anonymous Referee 1** for very valuable comments contributing to the improvement of the paper.

Specific comments
The authors should reconsider the importance/necessity of the comparison with the Lakagígar eruption. Both eruptions are completely different (latitude, date, vicinity to Czech lands. . .) so the authors should clearly explain why it is interesting this comparison between them. Moreover, it is important that the authors explain clearly the different features of the two eruptions. In this point, I think that another option is focusing the paper only in the Tambora eruption.
RE: The paper is newly oriented only on the Tambora eruption. Everything related to Lakagígar eruption was deleted and parts related to Tambora were changed accordingly.

The discussion section is not clearly linked with the result sections; this is more evident in sections 5.2 and 5.3
RE: Because of excluding parts of the manuscript related to the Lakagígar eruptions, we changed discussion and we hope to be more close to the results presented.

One of the main conclusions of the paper is that the Tambora eruption impacted less in the climate and more in the society that the Lakagígar one. But I miss a discussion about why this happened.
RE: Because of deleting effects of the Lakagígar eruption, these effects of both eruptions are not directly compared. Climatic and human impacts of the Tambora eruption then follow from corrected results as well as corrected discussion.

Introduction The unidentified eruption of 1809 is cited in the introduction. But nothing about this eruption is explained in the rest of the text. This eruption can affect the short-term analysis presented in the paper because "the mean temperature for each month was calculated using temperature data from five years prior to the eruption", some discussion about that could be interesting.
RE: We add some related sentences into the last paragraph of Section 5.1: "In the light of papers by Cole-Dai et al. (2009) and Guevara-Murua et al. (2014), the cold summers early in the second decade of the 19th century may also have been influenced by an unknown volcanic eruption in 1808/1809. In this context, Brönnimann (2015) demonstrated cool April–September 2010 patterns compared to mean surface air temperatures in 1801–1830 and argued that this eruption could have set the stage for sustained ocean cooling (compare Stenchikov et al., 2009). However, 1811 was already warmer in the Czech Lands from spring to autumn, and lower temperatures started in 1812 (see Fig. 2)."

About the impact of Lakagígar out of Europe could be interesting to cite Trigo et al (2010). Also could be useful in the discussion about the foggy events. Ordering the archival sources the S1 must be cited the first in the text then S2.

RE: Because of reduction of the manuscript only on the Tambora eruption, Trigo et al. (2010) was not included into References.

Methods No methods are described for the use of the documentary data (no instrumental).
RE: Accepted, the new paragraph related to the use of documentary (instrumental) data was added as follows: "In this paper, descriptions of weather and related phenomena in the Czech Lands post-Tambora, i.e. May 1815–December 1817 are derived from documentary data. All such the data extracted were critically evaluated, including analysis of source credibility, place and time attribution of records, content analysis, interpretation of records with respect to recent meteorological terminology and cross-checking of records against various different places in the Czech Lands. The creation of a database was the next step, in which information about place, time and event, characterised by key-words, full reports and data sources, has been recorded to provide a basis for further use (see Section 4.2). Kreybich's records from Žitenice (S1–S3) and Hausner's observations from Buchlovice (S4) were then further employed for calculation of monthly numbers of precipitation days in 1815–1817 (see Fig. 6).

The climatic effects of the volcanic eruption based on instrumental observations are expressed in the short-term and long-term contexts. In the short-term, the approach followed is that taken by several other papers addressing the effects of eruptions on temperature series (e.g. Sear et al., 1987; Robock and Mao, 1995; Kelly et al., 1996; Písek and Brázdil, 2006; Fischer et al., 2007). Temperature patterns related to the eruption are described over a ten-year period to avoid the possible influence of a strong trend. The month of the eruption is taken as month zero. The mean temperature for each month was calculated using temperature data from five years prior to the eruption. Each monthly mean temperature for five years before and after the eruption was then expressed as a departure from the calculated mean value. The same approach was applied to series of precipitation totals. For the long-term context, the eruption year and two subsequent years were characterised by their order and magnitude in the whole series shown in increasing (temperatures) or decreasing (precipitation) order."

Results
Pag. 3 line 33-37 This paragraph would be better in the introduction with a comparison with the Lakagígar eruption. I do not like the structure. I think that some information given in "Post-volcanic weather and impacts on society" are "climatic responses". I propose a year by year structure but with all the information (instrumental and documentary, climatic and social) for each year.
RE: Accepted. The corresponding paragraph was included on the beginning of the second paragraph in Introduction: "A great deal of literature has been devoted to analysis of the climatological and environmental effects of the Tambora eruption. The volcanic eruption of Tambora (Lesser Sunda Islands, Indonesia) in April 1815, is among the most powerful of its kind recorded, classified at an intensity of 7 in terms of Volcanic Explosivity Index (VEI) (a relative measure of volcanic explosiveness, VEI is an open-ended scale that ranges from 0 to 8, where 8 represents the most colossal events in history. It is based on the amount of volcanic material ejected and the altitude it reaches – see Newhall and Senf, 1982)."
Concerning of joining of instrumental and documentary data year by year we do not see as too useful with respect to different suite of data. We mentioned it inn introductory paragraph to Session4 as: "This section describes climate, weather and related phenomena in the Czech Lands during the time after the Tambora eruption. Because the character of the data differs quite

sharply, a division is maintained between information obtained from quantitative meteorological measurements and more qualitative data arising out of documentary evidence."

Pag 4. Line 29. When are the haymaking and the grain harvest?
RE: Haymaking is in average running before the mid-June and grain harvest in the third decade of July. But in this context we only say that haymaking and the grain harvest have run during the rainy weather, i.e. in bad weather conditions. Because we speak before about summer months, attribution both activities to summer is apparent.

Page 4 line 29-30 "if two days were fine, it then rained for two days." This phrase it is not clear for me, is it referred to august?.
RE: This sentence follows after mentioning of August, i.e. this concerns of August. We hope that change of phrase on "if two days were fine, it then rained for following two days" is better understandable. It means that any days of fine weather were immediately followed by rainy weather.

Page 4 line 30 "The wine vintage was bad for the third year" I do not understand this phrase, what year is the third? 1815? Is there some climatic explanation for the caterpillars plague in May?
RE: Accepted and corrected as: "The wine vintage of 1815 was bad for the third year, after 1813 and 1814 (S4)." Sorry there is not any climatic explanations for caterpilars. We put this sentence with respect to the fact that it had influence on bad harvest of fruits which were important part of nutrition for people.

Pag. 4 line 36-37. "Kreybich reports a flood on the Elbe for 10–14 August with extensive damage to agricultural crops" is it known the specific location? Zitenice?
RE: Accepted and corrected as: "In a similar vein, Kreybich in his records at Žitenice reports a flood on the Elbe for 10–14 August with extensive damage to agricultural crops (S1)."

Pag 4. Line 41. The dry autumn of 1815 is also clear identified in figure 4.
RE: Accepted, the corresponding sentence was changed as follows: "The wet, cold summer gave way at the end of August to a very dry, cold autumn in 1815, confirmed by sources from Bohemia (S1) and Moravia (S4), and clearly documented by negative precipitation anomaly (Fig. 5) and lower monthly numbers of precipitation days (Fig. 6)."

Pag 5 line 11 "Other Czech documentary sources report 1816 as particularly cold and wet, with bad harvests and rising prices of all products" this phrase need a cite.
RE: This sentence introduces several documentary data, which follows afterwards. Making clearer this context, we changed subsequent sentence as follows: "For example, around Nové Město na Moravě …"

Pag 6 line 11 "shortages" of what? food? water?
RE: Accepted and corrected as: "shortages of food"

Pag. 6 25-29. I see better this paragraph in the introduction and developing a comparison with the Lakagígar eruption.

RE: This paragraph was deleted with respect to restriction of the paper only to the Tambora eruption.

Pag. 7 Many references to thunderstorms during the Lakagígar eruption but also during the Tambora. Can you discuss deeply how this phenomenon could be induced by the eruptions?.
RE: Removing the Lakagígar eruption from the article, not any particular thunderstorms are reported, i.e. proposed discussion would be not relevant.

Figures
Figure 1: It would be interesting including a legend to explain which locations have instrumental information (temperature and precipitation) and/or documentary information.
RE: Corrected as requested.

Figure 2, 3, 4: Does it make sense including the Chez Lands series? This series during this period is calculated from Prague and Brno. Both included in the figures.
RE: We see including of the Czech series as useful. In the period analysed it is not only simple average of the two series because of method of calculation used (both series were adjusted with respect to 1961–2000 temperature patterns – see Brázdil et al., 2012a).

Figure 6: Redundant, all the information in this figure is also in figure 10.
RE: Figure 10 was deleted.

Technical comments Be coherent with format of the dates 7 April or 28th April.
RE: This concerns of formulations "between 11th and 28th April" and "between 17th and 28th April", otherwise we use the first type of writing. We consulted it with a native speaker: if we use only "between 17 and 28 April", it implies that it snowed only once and we don't know when. From this reason we let it in its original form.

We would like to thank **Ricardo Trigo (referee 2)** for very valuable comments contributing to the improvement of the paper.

**1. Major comments**

**1.1. (Novelty of datasets used and results obtained)** It is not entirely clear to readers the level of novelty of the various datasets presented in section 2.2. If I understood correctly all datasets have been described/used in the past, with the exceptions of the documents related to Reverend Simon Hausner and the teacher Noviny pod Ralskem. This is important to understand if the authors have simply used datasets compiled previously (even if often by themselves) or if new datasets where explored within the scope of this particular work. **Please clarify.**

RE: The sections 2.1 and 2.2 related to data was newly re-elaborated. Using of all data available for the analysis of the Tambora eruption is new because we did not yet worked with this topic. Some of these datasets were already reported or elaborated in some other publications (i.e. in referee's understanding "dataset compiled previously"), but with other aims than in this topic. Other data were not yet published, but were extracted during the systematic historical-climatological research running for couple of years in our institute. The new paragraphs 2.1 and 2.2 were changed as follows:

**"2.1 Instrumental data**

The climatological analysis herein is based on the following monthly, seasonal and annual temperature and precipitation series for the Czech Lands (Fig. 1):

(i) Prague-Klementinum (central Bohemia): homogenised series of temperatures (1775–2010) and precipitation (1804–2010), starting in a block of buildings that were once the Jesuit college of St. Clement, and located on the same site until quite recently (for data see Brázdil et al., 2012a)

(ii) Brno (south-eastern Moravia): homogenised series of temperatures (1800–2010) and precipitation (1803–2010) compiled from a number of places in the Brno area and homogenised to the recent Brno airport station (for data see Brázdil et al., 2012a)

(iii) Czech Lands: series of mean areal temperatures (1800–2010) and mean areal precipitation (1804–2010) calculated from ten homogenised temperature series and 14 homogenised precipitation series over the Czech Lands (for data and details of calculation, see Brázdil et al., 2012a, 2012b)

(iv) Žitenice (north-western Bohemia): homogenised series of temperatures (1801–1829) measured by parish priest František Jindřich Jakub Kreybich at Žitenice (measurements started in 1787 but incomplete before 1801), worked up by Brázdil et al. (2007)

(v) Central Europe: reconstructed temperature series (AD 1500–2007), consisting of temperatures derived from documentary-based temperature indices for Germany, Switzerland and the Czech Lands up to 1759 and homogenised temperature series of 11 secular meteorological stations located in these three countries and Austria from 1760 onwards (Dobrovolný et al., 2010).

**2.2 Documentary data**

The pre-instrumental and early-instrumental period of meteorological observations in the Czech Lands is well covered by documentary evidence that contains information about weather and related phenomena. It occurs in a number of data sources (e.g. annals, chronicles, memoirs, diaries, newspapers, financial records, songs, letters, epigraphic records, and others), which provide the basis for research in historical climatology (Brázdil et al., 2005b, 2010b). As well as a wealth of chronicles and personal histories reporting various climatic and weather anomalies,

their impacts and consequences (for those used in this study see Section 4.2), the following sources have proved particularly valuable:

(i) Annual summaries of the weather and the general economic situation that accompany the daily weather observations kept by František Jindřich Jakub Kreybich in Žitenice for the years 1815, 1816 and 1817 (S1–S3)

(ii) Qualitative daily weather observations and their monthly and annual summaries kept by Reverend Šimon Hausner of Buchlovice (south-eastern Moravia), spanning the 1803–1831 period (S4)

(iii) The detailed weather records kept by Anton Lehmann, a teacher in Noviny pod Ralskem, over the 1756–1818 period, which were copied into the local "book of memory" by Joseph Meissner in 1842 (S6)

(iv) Notes extracted from meteorological observations kept by Antonín Strnad and Alois David, the third and fourth directors of the Prague-Klementinum observatory (Poznámky, 1977).

Moreover, the editions of newspapers published in Prague (*Prager Zeitung*), Brno (*Brünner Zeitung*) and Vienna (*Wiener Zeitung*) covering the post-Tambora years were also systematically scrutinised for 1815–1817. Although weather information appears relatively rarely in their pages with respect to descriptions of events in the Czech Lands or Austria, related stories from other parts of Europe or North America clearly prevail there."

Moreover, to ensure reproducibility and homogenization of derived datasets it is common for authors to provide all methodological steps on the information and time series derived from documentary sources. Here no such information is provided in section 3 (Methods), underlining perhaps that these are not new datasets (?). **Please clarify.**

RE: Accepted, the new paragraph related to the use of documentary data was added as follows: "In this paper, descriptions of weather and related phenomena in the Czech Lands post-Tambora, i.e. May 1815–December 1817 are derived from documentary data. All such the data extracted were critically evaluated, including analysis of source credibility, place and time attribution of records, content analysis, interpretation of records with respect to recent meteorological terminology and cross-checking of records against various different places in the Czech Lands. The creation of a database was the next step, in which information about place, time and event, characterised by key-words, full reports and data sources, has been recorded to provide a basis for further use (see Section 4.2). Kreybich's records from Žitenice (S1–S3) and Hausner's observations from Buchlovice (S4) were then further employed for calculation of monthly numbers of precipitation days in 1815–1817 (see Fig. 6)."

It is clear that the authors have a large experience in past-climate analysis, particularly over Czech Republic. Thus, it is expected that all relevant literature for the main topic of this work (i.e. impacts of major eruptions in Czech lands) is provided at the introduction, allowing to stress the novelties that will be investigated here. Thus it is rather strange that the first time a key reference evaluating the impact of major eruptions in the mean Czech temperature region is mentioned only at the end (Page 11), and not in the introduction (Mikšovský et al, 2014). **Please clarify.**

RE: The new paragraph related to effects of volcanic eruption in the Czech Lands was added in Introduction part as follows: "There are only a few studies that address the effects of volcanic activity on the Czech Lands (central Europe). For example, Kyncl et al. (1990) analysed climatic reactions and tree-ring responses to the Katmai eruption (Alaska) in 1912, largely on a central European scale. Brůžek (1992) studied the impacts of large 19th–20th-century volcanic eruptions

upon temperature series at the Prague-Klementinum station. Brázdil et al. (2003) described a number of extreme climatic anomalies following the 1783 Lakagígar eruption (Iceland) in the course of an analysis of daily weather records covering 1780–1789, kept by Karel Bernard Hein in Hodonice, south-west Moravia. Písek and Brázdil (2006) used temperature records from Prague-Klementinum, together with other central European series (Kremsmünster, Vienna-Hohe Warte and Germany), to address the temperature effects of seven large tropical eruptions and nine eruptions in Iceland and the Mediterranean, complemented by short descriptions of the Lakagígar 1783 and Tambora 1815 events based on documentary data. This paper also included the effects of three tropical eruptions on series of sums of global radiation for the Hradec Králové station (together with Potsdam in Germany and Skalnaté Pleso in Slovakia). Brázdil et al. (2010) analysed climate and floods in the first post-Lakagígar winter (1783/1784) with particular reference to central Europe. Volcanic forcing was also taken into account as part of an attribution analysis of Czech temperature and precipitation series by Mikšovský et al. (2014) and in Czech series of spring and summer droughts by Brázdil et al. (2015b)."

**1.2. (Lack of statistical significance inference of several results)**. There are a number of interesting results describing weather/climate extremes that may be associated to the effects of both eruptions in the climate of the Czech Lands. However, many times the descriptions are not accompanied by a more robust statement on the statistical significance (or uniqueness) of the so called-extreme event. A few examples of that are highlighted here:
RE: Looking on the character of documentary evidence, when, in many cases, we are not able to create any series of quantitative values, it is very difficult to say, how the event was unique or what is his statistical significance. We are able only to say, that it was sure any extreme event (looking also on experience of contemporaries) which was worthy of interest and from this reason it was recorded, i.e. for memory of people for the future.

a) (Page 4, lines 33-36): "A message from Litoměřice dated 9 August reports a flood lasting eight days on the River Elbe after five weeks of rainy periods. The water rose to a level of two feet [c. 65 cm] under the bridge, so the structure survived, but grain, vegetable and other field crops were damaged (Katzerowsky, 1895)." **How exceptional is this situation? How many time has it occurred in the last 300 years?**
RE: The problem is that documentary data usually give for some particular place incomplete data with emphasis on realy extreme events. From this point of view we are sure that it was severe flood when water went more than half a meter above the bridge. But to say, for example, how many times it occurred during the past 300 years (it is to calculate any N-year re-occurrence period) is practically impossible. Moreover, systematic water-level measurements started at Litoměřice since 1851, it is any more exact comparison is not available.

b) (Page 5, lines 8-10): "The ice was definitely gone by 8–9 March (S2). Lehmann reports a 3/4-ell [c. 58-cm]-thick crust of ice on some fields in Noviny pod Ralskem (S6). Frosty weather prevailed in March with blizzards from 26 to 31 March. April was cold and dry, with no heavy rain (S4)." **Again, to what extent are these descriptions unique in the longer term context?**
RE: We are just describing weather course or interesting weather (climate) anomalies/events which occurred after the Tambora eruption and which we know from instrumental records and documentary evidence. This means that we are not reporting if every such message is unique in the long-term context or not. Where it is possible, we are trying to explain it (see e.g. your point

c) below) what is, for example, question of temperature and precipitation anomalies (see Section 5.1).

c) (Page 6, lines 5-8): Kreybich, the Žitenice cleric, reports four landslides in spring, the result of extremely wet conditions in north-western Bohemia: the first on Křížová hora Mt. north of Žitenice, the second on Trojhora Hill between Chudoslavice and Třebušín, the third at Vitín near Malé Březno (community now defunct) and the fourth east of Jílové (S3). **Are landslides very rare in the area? How often do these occur?**

RE: Following text, characterising landslides described, was added: " Five landslides in 1817 in north-western Bohemia, which are not included in the historical catalogue of landslides by Špůrek (1972), are the three most important events of this kind to appear in documentary evidence before 1900. Other recorded documented landslides in this area took place only in 1770, as a result of the very wet and rainy year of 1769, and in winter 1769/1770 (see e.g. Raška et al., 2016) and in 1897–1900, due to persistent wet and rainy patterns (Rybář and Suchý, 2000)."

**1.3. (The choice of Tambora vs Lakagigar is not clear**). It is not clear to readers the choice of these two eruptions that are so different in their characteristics, location, impacts, etc. A more straightforward approach would be to consider several major tropical explosive eruptions (as those listed in Fischer et al. 2007) or, alternatively, major eruptions in high latitudes (particularly in Iceland). Besides taking place roughly with 30 years apart, it is not entirely clear the rationale for the combined assessment. **Please clarify.**

RE: The paper is reduced only on Tambora eruption and its consequences. Parts related to Lakagígar were deleted.

Please notice that the differences between the two types of eruptions are so large that they have implications in the literature cited (that can be quite different) and even way their impact is evaluated. In particular the definition of month 0 (and in fact year 0, 1 and 2) is quite unclear to me in the case of the eruption of Lakagigar that took place between (1783 and early 1784). **Please clarify**

RE: Part of the manuscript related to Lakagígar was deleted.

**2. Minor suggestions/comments**
**2.1.** (Page 3, sections methods) Please provide 1 or 2 references to support the various options explained, particularly the 5+5 years used before and after the eruption.

RE: Accepted, we add following sentence with some quotations: "The climatic effects of the volcanic eruption based on instrumental observations are expressed in the short-term and long-term contexts. In the short-term, the approach followed is that taken by several other papers addressing the effects of eruptions on temperature series (e.g. Sear et al., 1987; Robock and Mao, 1995; Kelly et al., 1996; Písek and Brázdil, 2006; Fischer et al., 2007)."

**2.2.** (Page 3, end of section 4.1) I think that this section would gain with a sentence explaining that major tropical eruptions (e.g. Tambora-1815, Krakatoa-1883, Pinatubo 1991) have the capacity to alter the radiative balance for the entire world, impinging widespread cooling at the surface level of the globe, but often inducing large-scale changes in the atmospheric circulation that can warm the continental areas in winter (see carefully Robock 2002, Science).

RE: Accepted. Because of the comment of the referee 1, we had to move small Section 4.1 to Introduction. In the first paragraph of Introduction we included two sentences, following your

request: "The effects of large tropical volcanic eruptions on radiative balance manifest themselves not only in widespread cooling, but also contribute to large-scale changes in atmospheric circulation, leading to one or two post-volcanic mild winters in the Northern Hemisphere (Robock, 2000). Fischer et al. (2007) associated volcanic activity with a positive phase in the North Atlantic Oscillation (NAO), causing stronger westerlies in Europe and wetter patterns in Northern Europe."

**2.3.** (Page 3, sections 4.1) The term VEI has not been described before. Please provide its meaning here when it appears for the first time (Volcanic Explosivity Index, VEI). It would be also useful to give a range of its scale between 1 and 8 (and a glimpse of the logarithmic nature of its scale, thus emphasizing the much larger volume of lava associated to a VEI-7 when compared to a VEI-6).
RE: Corrected. Explaining text related to VEI was added as follows: " The volcanic eruption of Tambora (Lesser Sunda Islands, Indonesia) in April 1815, is among the most powerful of its kind recorded, classified at an intensity of 7 in terms of Volcanic Explosivity Index (VEI) (a relative measure of volcanic explosiveness, VEI is an open-ended scale that ranges from 0 to 8, where 8 represents the most colossal events in history. It is based on the amount of volcanic material ejected and the altitude it reaches – see Newhall and Senf, 1982)."

**2.4.** (Page 4, lines 1-2, Fig. 2) The 5 lines used in Fig.2 are very similar and it is not clear the exception mentioned for Brno as being particularly milder than the others for the winter 1816/1817 (?)
RE: Accepted. Because it is not clearly visible in Fig. 2, the formulation were modified to make this point more clear: "After a very mild winter of 1816/1817 (the mildest in the 1811–1820 period in four series; only winter in Brno 1814/1815 was slightly warmer), negative anomalies occurred, especially in spring with the strongest negative anomaly (stronger than in summer 1816)."

**2.5.** (Page 4, Section "The year 1815") Are the author implying that the "cold May 1815 with more frequent rain and frosts on 29–30 May" are related to the Tambora eruption? And the same doubt applies to the reference to the fruit trees eaten by caterpillar.
RE: We are not implying, that any events described are direct effect of Tambora eruption. We are just describing weather or interesting weather (climate) anomalies/events which occurred after the Tambora eruption and which we know from instrumental records and documentary evidence. Sentence about "eaten fruit trees by catterpilar" is included because it had influence on bad harvest of fruits which were important part of nutrition for people.

**2.6.** (Page 4, Section "The year 1815") Are the authors implying that the "cold May 1815 with more frequent rain and frosts on 29–30 May" are related to the Tambora eruption?
RE: See response to the previous point 2.5.

**2.7.** (Page 6, line 17) Please provide a reference to Fig.6 earlier at the end of the sentence: "…driving prices up from 1813 onwards, culminating in 1817 (Fig. 6)".
RE: Corrected as requested.

**2.8.** (Page 7, lines 33-38) Several specific extreme weather events are mentioned here (e.g. March 1784; April 1785). A number of works for other sectors of Europe have been developed for the

years post-Lakagigar, please provide some links to these works in terms of compatibility (or not) of the atmospheric circulation anomalies.

RE: Parts of the manuscript related to Lakagígar eruption were completely deleted.

**2.9.** (Page 8, section 5.2) The contents of this section are not particularly well incorporated into the overall flow of the text. First, this discussion is not structured with Tambora being analysed after Lakagigar (that should be probably the most natural order, but the authors have preferred the reverse from the beginning). Secondly the temporal and spatial link between these various theories (earthquake in Messina 1783, Comet in 1811, Number of sunspots in 1814, etc) is not provided in a meaningful way.

RE: Corrected as requested. A part related to Lakagígar was deleted and a part related to Tambora was changed accordingly.

**Figures**

Fig1 Please provide different symbols for stations with different information. For example Prague and Brno should have a distinct symbol. The same apply for those locations with just documentary sources.

RE: Corrected as requested.

Fig3. I believe the figure caption should read: "Difference between mean summer and winter…"

RE: Corrected as requested.

Fig. 6 It seems that the contents of this figure is repeated in Fig 10 (?)

RE: Figure 10 was deleted.

Fig. 7 I believe that the time delimitation of Lakagigar eruption should extend until February 1784.

RE: This figure was deleted.

We would like to thank **Anonymous Referee 3** for very valuable comments contributing to the improvement of the paper.

1. There might be some problem in this paper's structure and contents. The authors have centered their paper on the 1815 eruption and its consequences. But they also include information about the Lakagigar eruption mainly within the "results" section (only a short paragraph in the introduction). This is quite confusing. Even the title is too long and quite ambiguous. Moreover the two eruptions are rather different and so are their impacts. I think the authors could consider two solutions
- A) Either concentrate on the 1815 eruption, as they possess more instrumental and documentary information
- B) Or write a paper on the comparison of the two eruptions and their consequences, change the title and modify the paper's structure accordingly, using and developing the texts where this comparison is already carried out.
RE: The option A was selected. The paper orients only on the Tambora eruption and the title was changed as follows: Climatic effects and impacts of the 1815 eruption of Mount Tambora in the Czech Lands

2. Introduction - Explain how an eruption in tropical site may affect the climate in central Europe. If you include the L. eruption, compare the features of both eruptions.
RE: Accepted. We believe that the recent improved version of the first paragraph is related also to central Europe, i.e. changes in the radiative balance are implying subsequent cooling and changes in circulation. See particular: "For example, Fischer et al. (2007) analysed winter and summer temperature signals in Europe following 15 major tropical volcanic eruptions and found significant summer cooling on a continental scale and somewhat drier conditions over central Europe. The effects of large tropical volcanic eruptions on radiative balance manifest themselves not only in widespread cooling, but also contribute to large-scale changes in atmospheric circulation, leading to one or two post-volcanic mild winters in the Northern Hemisphere (Robock, 2000). Fischer et al. (2007) associated volcanic activity with a positive phase in the North Atlantic Oscillation (NAO), causing stronger westerlies in Europe and wetter patterns in Northern Europe. Literature addressing volcanic effects on precipitation is more sparse (Gillett et al., 2004). For example, Wegmann et al. (2014) analysed 14 tropical eruptions and found an increase of summer precipitation in south-central Europe and a reduction of the Asian and African summer monsoons in first post-eruption years. Weaker monsoon circulations attenuate the northern element of the Hadley Cell and influence atmospheric circulation over the Atlantic-European sector, contributing to higher precipitation totals."

3. Data section – Documentary data and the notes that accompany some of the instrumental data should be described in more detail. In some cases, the authors refer to their own past publications, but a short sentence could clarify the content of each of the sources (e.g. 1), p. 3, l.11). Indicate clearly the new information brought about by this paper.
RE: Accepted, information about instrumental and documentary data were re-elaborated – see new Sections 2.1 and 2.2 below:
**"2.1 Instrumental data**
The climatological analysis herein is based on the following monthly, seasonal and annual temperature and precipitation series for the Czech Lands (Fig. 1):

(i) Prague-Klementinum (central Bohemia): homogenised series of temperatures (1775–2010) and precipitation (1804–2010), starting in a block of buildings that were once the Jesuit college of St. Clement, and located on the same site until quite recently (for data see Brázdil et al., 2012a)

(ii) Brno (south-eastern Moravia): homogenised series of temperatures (1800–2010) and precipitation (1803–2010) compiled from a number of places in the Brno area and homogenised to the recent Brno airport station (for data see Brázdil et al., 2012a)

(iii) Czech Lands: series of mean areal temperatures (1800–2010) and mean areal precipitation (1804–2010) calculated from ten homogenised temperature series and 14 homogenised precipitation series over the Czech Lands (for data and details of calculation, see Brázdil et al., 2012a, 2012b)

(iv) Žitenice (north-western Bohemia): homogenised series of temperatures (1801–1829) measured by parish priest František Jindřich Jakub Kreybich at Žitenice (measurements started in 1787 but incomplete before 1801), worked up by Brázdil et al. (2007)

(v) Central Europe: reconstructed temperature series (AD 1500–2007), consisting of temperatures derived from documentary-based temperature indices for Germany, Switzerland and the Czech Lands up to 1759 and homogenised temperature series of 11 secular meteorological stations located in these three countries and Austria from 1760 onwards (Dobrovolný et al., 2010).

**2.2 Documentary data**

The pre-instrumental and early-instrumental period of meteorological observations in the Czech Lands is well covered by documentary evidence that contains information about weather and related phenomena. It occurs in a number of data sources (e.g. annals, chronicles, memoirs, diaries, newspapers, financial records, songs, letters, epigraphic records, and others), which provide the basis for research in historical climatology (Brázdil et al., 2005b, 2010b). As well as a wealth of chronicles and personal histories reporting various climatic and weather anomalies, their impacts and consequences (for those used in this study see Section 4.2), the following sources have proved particularly valuable:

(i) Annual summaries of the weather and the general economic situation that accompany the daily weather observations kept by František Jindřich Jakub Kreybich in Žitenice for the years 1815, 1816 and 1817 (S1–S3)

(ii) Qualitative daily weather observations and their monthly and annual summaries kept by Reverend Šimon Hausner of Buchlovice (south-eastern Moravia), spanning the 1803–1831 period (S4)

(iii) The detailed weather records kept by Anton Lehmann, a teacher in Noviny pod Ralskem, over the 1756–1818 period, which were copied into the local "book of memory" by Joseph Meissner in 1842 (S6)

(iv) Notes extracted from meteorological observations kept by Antonín Strnad and Alois David, the third and fourth directors of the Prague-Klementinum observatory (Poznámky, 1977).

Moreover, the editions of newspapers published in Prague (*Prager Zeitung*), Brno (*Brünner Zeitung*) and Vienna (*Wiener Zeitung*) covering the post-Tambora years were also systematically scrutinised for 1815–1817. Although weather information appears relatively rarely in their pages with respect to descriptions of events in the Czech Lands or Austria, related stories from other parts of Europe or North America clearly prevail there."

4. The methods section (p. 3) should be more clear and developed, particularly when it comes to documentary data (different steps that were necessary to construct a dataset from the documentary data). This is included in other papers from the same authors but should be incorporated here referring to these specific cases.

RE: Accepted, the new paragraph related to the use of documentary data was added as follows: "In this paper, descriptions of weather and related phenomena in the Czech Lands post-Tambora, i.e. May 1815–December 1817 are derived from documentary data. All such the data extracted were critically evaluated, including analysis of source credibility, place and time attribution of records, content analysis, interpretation of records with respect to recent meteorological terminology and cross-checking of records against various different places in the Czech Lands. The creation of a database was the next step, in which information about place, time and event, characterised by key-words, full reports and data sources, has been recorded to provide a basis for further use (see Section 4.2). Kreybich's records from Žitenice (S1–S3) and Hausner's observations from Buchlovice (S4) were then further employed for calculation of monthly numbers of precipitation days in 1815–1817 (see Fig. 6)."

5. The results sections (p. 3- 7) should be reorganised according to your choice of A) or B) (see above, please). Should not the comparison of the two eruptions referring to climate and to their impacts be included in the results part? (if you follow B. If you select A, then these paragraphs should be deleted).

RE: Accepted. Because of selection of option A, this paragraph was deleted.

6. Rewrite the discussion part adding some current explanations about the differences of the two eruptions and why are the impacts different (if you choose B), putting the events into European context.

RE: Accepted. With respect to change only on the Tambora eruption, everything in the manuscript related to Lakagígar eruption was deleted and parts related to Tambora were changed accordingly.

p. 3, line 9 –Explain what are visual weather records

RE: Under "visual weather records" we understand more-or-less systematic daily weather observations done without any instruments on the qualitative way. But due to restriction of the article only to the Tambora eruption, corresponding sentence was deleted.

p.4, 2nd paragraph. As the authors notice there had been already a cool period in 1812-1814. Perhaps the authors should point out more clearly the differences between these two cold periods and the drop of temperature anomaly after 1815.

RE: Accepted, we add following sentences into Section Discussion where it seems more appropriate than in the results as proposed: "In the light of papers by Cole-Dai et al. (2009) and Guevara-Murua et al. (2014), the cold summers early in the second decade of the 19th century may also have been influenced by an unknown volcanic eruption in 1808/1809. In this context, Brönnimann (2015) demonstrated cool April–September 2010 patterns compared to mean surface air temperatures in 1801–1830 and argued that this eruption could have set the stage for sustained ocean cooling (compare Stenchikov et al., 2009). However, 1811 was already warmer in the Czech Lands from spring to autumn, and lower temperatures started in 1812 (see Fig. 2)."

P. 4, line 16 (and Fig.4) – you refer that autumn 1817 shows strong negative anomalies, but autumn 1815 has also little rain. Please explain.

RE: Analysis of precipitation data after post-Tambora years allows us to detect some climatic anomalies which are clearly reflection of some circulation patterns (e.g. more often high-pressure situations over Central Europe in the both mentioned autumns). But we are not able to go behind because we do not know papers explaining circulation changes after large tropical volcanic eruptions in Europe what is not intention of this article.

Figure 1 – indicate through different symbols the places from where you used meteorological and documentary data. If you have both data for the same site use a combined symbol.

RE: Corrected as requested.

Figure 2 – why are the anomalies calculated relatively to the five years' period preeruption?

RE: We used method of analysis applied in several papers dealing with effects of volcanic eruptions in temperature series: "The climatic effects of the volcanic eruption based on instrumental observations are expressed in the short-term and long-term contexts. In the short-term, the approach followed is that taken by several other papers addressing the effects of eruptions on temperature series (e.g. Sear et al., 1987; Robock and Mao, 1995; Kelly et al., 1996; Písek and Brázdil, 2006; Fischer et al., 2007)." As mentioned in Section 3: "Temperature patterns related to the eruption are described over a ten-year period to avoid the possible influence of a strong trend."

Figure 3- The Figure caption is not clear and the two "temperature _C" in the vertical axes are confusing. You could write in the right one "Temperature anomalies in C.E." and leave the left one as it is.

RE: Corrected as requested.

Figures 6 and 10 – there is no need to include both figures.

RE: Accepted, Figure 10 was deleted.

We would like to thank **Anonymous Referee 4** for very valuable comments contributing to the improvement of the paper.

Review:
The authors compare the effects of two major volcanic eruptions, the ones of Laki and Tambora, on climate in central Europe. The paper is interesting in that it provides both climate information, the historical background, and climate impacts information (e.g., as measured in prices). In this sense it is a rich paper that fills an important gap, and the discussion has some interesting elements. It is of course timely (with the bicentenary of the "Year Without a Summer" of 1816 and it is well written.
However, in terms of science the paper has some weaknesses that need to be addressed. I therefore suggest revisions as detailed below.
We would like to thank the referee 4 for very valuable comments contributing to the improvement of the paper. Before replying to your very valuable comments we have to stress, that referees 1-3 have been so strongly against joining the both Laki and Tambora eruptions in one article that we had to skip everything related to Laki and the revised paper concerns only of Tambora under new title: Climatic effects and impacts of the 1815 eruption of Mount Tambora in the Czech Lands.

Major comments
1. My man concern is that climatic anomalies might be unduly attributed to volcanic eruptions. Not every climatic anomaly immediately following Tambora or Laki is volcanically caused. Arguably the largest contribution to the "Year Without a Summer" of 1816 was random internal variability. Some further comments on that follow below.
RE: We are just describing weather course or interesting weather (climate) anomalies/events which occurred after the Tambora eruption and which we know from instrumental records and documentary evidence. We are not saying that any of climatic anomalies described would be any direct effect of Tambora eruption.

2. Tambora effects are described already for June 1815 (e.g., p. 4, l. 15). That's around the tie it takes to form aerosols from the sulphur, so by early June the aerosols just about start to affect the tropics. This is an instance where the results should be discussed better in the context of the mechanisms and expectations.
RE: Please look on our previous explanations. We are describing this situation just as post-Tambora time.

A second concern is the lack of a clear description of the mechanisms. The mechanisms behind the climate effects of Laki and Tambora arguably differ a lot (on Laki see work by Luke Oman, of Highwood and Stevensen, Schmidt et al., etc. see below; for Tambora there are myriads of papers). This not reflected adequately in the paper.
RE: Effects or mechanisms of climate effects after Tambora (tropical eruption) are reported in the first paragraph of the introduction as:
"Violent tropical volcanic eruptions, transporting large quantities of particles into the lower stratosphere, give rise to decreases in temperatures in the troposphere, which cools for two or three subsequent years in response to strongly enhanced back-scattering of incoming solar radiation (Robock and Mao, 1995; Briffa et al., 1998; Robock, 2000; Jones et al., 2004; Písek and

Brázdil, 2006; Timmreck, 2012; Lacis, 2015; LeGrande and Anchukaitis, 2015). Camuffo and Enzi (1995) studied the occurrence of clouds of volcanic aerosols in Italy over the past seven centuries with particular attention to the accompanying effect of "dry fog". Volcanic cooling effects are best expressed in temperature series averaged for a large area after significant tropical volcanic eruptions (Sear et al., 1987; Bradley, 1988; Briffa et al., 1998; Sigl et al., 2015). For example, Fischer et al. (2007) analysed winter and summer temperature signals in Europe following 15 major tropical volcanic eruptions and found significant summer cooling on a continental scale and somewhat drier conditions over central Europe. The effects of large tropical volcanic eruptions on radiative balance manifest themselves not only in widespread cooling, but also contribute to large-scale changes in atmospheric circulation, leading to one or two post-volcanic mild winters in the Northern Hemisphere (Robock, 2000). Fischer et al. (2007) associated volcanic activity with a positive phase in the North Atlantic Oscillation (NAO), causing stronger westerlies in Europe and wetter patterns in Northern Europe. Literature addressing volcanic effects on precipitation is more sparse (Gillett et al., 2004). For example, Wegmann et al. (2014) analysed 14 tropical eruptions and found an increase of summer precipitation in south-central Europe and a reduction of the Asian and African summer monsoons in first post-eruption years. Weaker monsoon circulations attenuate the northern element of the Hadley Cell and influence atmospheric circulation over the Atlantic-European sector, contributing to higher precipitation totals."

Because they are generally known we suppose that further extension is not necessary (moreover, we would be only able to repeat, what was already said in many published papers because we are not coming with any new ideas in this field).

At some point in the paper, e.g., at the start of Section 5, the expectations should be stated. Tropical eruptions have a direct radiative effect that is stronger in summer than in winter, then there are indirect effects that affect the circulation over the North Atlantic and Europe in late winter and perhaps other effects that operate in summer. A high latitude eruption has a direct effect that may be stronger if the eruption is sustained long enough. There may also be indirect effects (see literature given below).
RE: Facts related to Tambora we mentioned in the first paragraph of Introduction (see previous point) and Laki was excluded from elaboration.

Personally I liked the discussion of the quite different political situation and historical background. Comparig 1783/4 and 1815-7 is certainly interesting from that point of view.
RE: Thank you for positive evaluation but it was not positively evaluated by referees 1-3, i.e. in the revised paper we limit it only to Tambora.

Minor
Abstract, l. 20: extremely cold and wet... really?
RE: Accepted and corrected as: "Czech documentary sources make no direct mention of the Tambora eruption, neither do they relate any particular weather phenomena to it, but they record extremely wet summer for 1815 and extremely cold summer for 1816 (the "Year Without a Summer") …"|

Abstract, l. 28-29: be careful in attribution
RE: Accepted, this sentence was cancelled in connection to the article reduction.

One of the important new aspects is the link to prices. However, the data on prices are not well explained. How was the market regulated? Do we need to take inflation into account, etc.

RE: We are sorry, but we only took series of prices from published papers, i.e. we did not make any own research which is also out of focus of this publication. By reading of papers (particularly those three from which we taken data, as well as other basic price-related references) we did not find any focussed expressions to prices in the 1810s, besides some general proclamations in relation to prices. This means, that we do not have any information about market regulation. Looking on cereal prices in 1811-1820, it seems that inflation was not playing any important role during this few pre-Tambora and post-Tambora years.

It might make sense to also compare with the HISTALP data.

RE: Because we limit our paper only to patterns in the Czech Lands, we do not mention directly HISTALP data. But in calculation of Central European temperature series (used in Section 4.1) 10 homogenised stations from HISTALP were used and they were applied also as reference series in homogenisation of long-term Czech series (Prague-Klementinum and Brno).

P. 3, L. 25: What is month zero for an 8-month eruption? The first?

RE: The Laki eruption was moved from the article.

P. 3, L. 39: The title "Climatic responses" already does an attribution. Better: Climatic anomalies...

RE: Accepted, the title was changed as: 4.1 The Tambora eruption in the context of meteorological observations

P. 5, l. 8: 58 cm? Sounds like a local phenomenon.

RE: Yes, this figure is related to a particular record, but it is in quantitative form. We see this information important because it says that there was a thick ice.

P. 5, l. 22: 191 rain days. This sort of information is very useful, but it should be accompanied by a norm. What do we expect in a normal year? (And over which period did it rain on 191 days? not clear)

RE: This information was complemented as follows: "It rained for 191 days of the year at Žitenice (S2); the mean for 1806–1818 is 166 days (Brázdil et al., 2007)." Time attribution follows from manuscript: it is included in the paragraph "The year 1816" and corresponding words are: "for 191 days of the year".

P. 6, l. 1: The cold spring 1817 is interesting.

RE: This follows from homogenised series – we do not have any particular explanations for it.

P. 6, l. 21: More information on the prices is needed.

RE: Please see our expression above to the problem of prices. We did some changes in this paragraph as: "The qualitatively-described increase in prices may be confirmed by actual records of mean prices for the basic grain crops. Data from Prague in Bohemia and for Moravia, indicate bad harvests in 1815 and 1816 driving prices up from 1813 onwards, culminating in 1817 (Fig. 7). While in Moravia grain prices rose threefold (doubling for oats), the figures for Prague were c. 4.5-fold for rye and barley and tripled for wheat. A higher increase in prices in Bohemia compared with Moravia has been confirmed for many other places in the province by Tlapák

(1977), but with prices available only up to 1817; for example, the figures for Litoměřice were fivefold for rye and barley and tripled for wheat and oats. Again the better harvest of 1817 drove prices down sharply, to the level of 1813 or below. While prices for wheat, rye and barley exhibited similar steep increases and decreases, fluctuations in those for oats were more stable, also due to a good yield in 1816 (S6)."

P. 6, l. 31: Again: Avoid attribution in the title
RE: This section including title related to Laki was cancelled.

P. 6, l. 32: Avoid starting a section with "Fig."
RE: This section related to Laki was cancelled.

P. 7, l. 15 and P. 8, . 33: The heavy thunderstorms without rain appear frequently in the historical descriptions. I personally do not see a physical reason for how this cold be related to an eruption. Going into any microphysical aspects is certainly well beyond this paper (I doubt that anybody would be able to do that).
RE: Quoting of thunderstorms was related to Laki – both were cancelled in the revised manuscript.

P. 9, l. 4ff: Interesting paragraph.
RE: Thanks for your evaluation.

P. 10, l. 15-26: Again, very interesting discussion.
RE: Thanks for your evaluation.

Figure 3: I do not understand the caption. Should it be: Differences between ..? And how is the value for JJA 1815 defined? Is it JJA 1815 minus Dec 1814 to Feb 1815 or JJA 1815 minus Dec 1815 to Feb 1816. In the former case, a Tambora signature is unexpected.
RE: Accepted, we add the word "between". This is calculated by standard (climatological) way as JJA 1815 minus DJF 1814/1815. We agree that in 1815 "Tambora signature is unexpected".

References
Highwood, E.-J. and Stevenson, D. S.: Atmospheric impact of the 1783-1784 Laki Eruption: Part II Climatic effect of sulphate aerosol, Atmos. Chem. Phys., 3, 1177-1189, doi:10.5194/acp-3-1177-2003, 2003.
Oman, L., A. Robock, G. Stenchikov, et al. 2006. "Modeling the distribution of the volcanic aerosol cloud from the 1783-1784 Laki eruption." J. Geophys. Res., 111 (D12): D12209
Oman, L., A. Robock, G. Stenchikov, G. A. Schmidt, and R. A. Ruedy. 2005. "Climatic response to high-latitude volcanic eruptions." J. Geophys. Res., 110 (D13): D13103
Schmidt, A., T. Thordarson, L. D. Oman, A. Robock, and S. Self. 2012. "Climatic impact of the long-lasting 1783 Laki eruption: Inapplicability of mass-independent sulfur isotopic composition measurements." J Geophys Res, 117 (D23): D23116
RE: Many thanks for these important references. Because Laki was excluded from the article, these references were finally not included.

---

## Author Response (AR2)

**Reviewer 2 - Responses**

I suggested to provide more quantitative evaluation of that exceptionality, that was achieved for the landslides example.

**RE**: Accepted by changes in the existing text and adding the new sentence as follows:
"The five landslides in 1817 in north-western Bohemia, which are not included in the historical catalogue of landslides by Špůrek (1972), are among the three most important landsliding events to appear in documentary evidence before 1900. Other recorded landslides documented in this area took place only in 1770 (14 landslides), as a result of the very wet and rainy year of 1769, and in winter 1769/1770 (see e.g. Raška et al., 2016) and in 1897–1900 (50 landslides altogether), due to persistent wet and rainy patterns (Rybář and Suchý, 2000). Apart from these three events, only 13 landslides in the remaining nine years during the 1770–1900 period are documented; this distribution also reflects the number of documentary sources available for extraction (Raška, 2016)."

However, in relation to the Floods in River Elbe you state that there is river flow data in Litomerice since 1851. If that is the case then you have 165 years of data, more than enough to compute the long-term return period of exceptional floods. Surely it is possible to classify the extreme event described in the sentence associated with that episode (?)

**RE**: Accepted by adding corresponding sentence estimating N-year return period to water level reported as follows:
"A message from Litoměřice dated 9 August reports a flood lasting eight days on the River Elbe after five weeks of rainy periods. The water rose to a level of two feet [$c$. 65 cm] under the bridge, so the structure survived, but grain, vegetable and other field crops were damaged (Katzerowsky, 1895). The water level reported would correspond to a $c$. 20-year return period if this were compared with systematically measured water levels at Litoměřice between 1851 and 1969 (Brázdil et al., 2005a)."